# Prenylcysteine oxidase 1 like protein is required for neutrophil bactericidal activities

Anastasiia Petenkova [1], Shelby A. Auger [2], Jeffrey Lamb[1], Daisy Quellier[1], Cody Carter[1], On Tak To[1], Jelena Milosevic[3], Rana Barghout[3], Abirami Kugadas[1], Xiaoxiao Lu[1], Jennifer Geddes-McAlister [4], Raina Fichorova[5], David B. Sykes [3], Mark D. Distefano [2] & Mihaela Gadjeva [1,6] ✉

The bactericidal function of neutrophils is dependent on a myriad of intrinsic and extrinsic stimuli. Using systems immunology approaches we identify microbiome- and infection-induced changes in neutrophils. We focus on investigating the Prenylcysteine oxidase 1 like (Pcyox1l) protein function. Murine and human Pcyox1l proteins share ninety four percent aminoacid homology revealing significant evolutionary conservation and implicating Pcyox1l in mediating important biological functions. Here we show that the loss of Pcyox1l protein results in significant reductions in the mevalonate pathway impacting autophagy and cellular viability under homeostatic conditions. Concurrently, *Pcyox1l* CRISPRed-out neutrophils exhibit deficient bactericidal properties. *Pcyox1l* knock-out mice demonstrate significant susceptibility to infection with the gram-negative pathogen *Psuedomonas aeruginosa* exemplified through increased neutrophil infiltrates, hemorrhaging, and reduced bactericidal functionality. Cumulatively, we ascribe a function to Pcyox1l protein in modulation of the prenylation pathway and suggest connections beween metabolic responses and neutrophil functionality.

*Pseudomonas aeruginosa* infections remain a serious clinical problem with an estimated 51,000 healthcare-associated infections in US hospitals each year. According to the CDC more than 6000 (13%) of these are multidrug-resistant, with about 440 annual deaths[1,2]. To define mechanisms that contribute to protection against *P. aeruginosa*, we compared resistance-associated signatures in neutrophils[3].

Previously, we reported that commensal organisms can influence neutrophil activation and, consequently, susceptibility to infection. Germ free (GF) Swiss Webster (SW) mice were more susceptible to *P. aeruginosa*-induced infection when compared to Specific Pathogen Free (SPF) SW mice[4]. This elevated susceptibility of GF SW mice to infection was rooted in decreased numbers and reduced PMN functionality[4]. GF-derived mature PMNs failed to kill *P. aeruginosa*, illustrating that both the quantity and quality of the responding PMNs depended on commensal presence[4]. Our observations were consistent with other studies examining changes in the bone marrow (BM) myeloid progenitor frequencies, neutrophil functionalities, and revealing that gut microbiota-derived signals affected granulopoiesis[5–7]. Gut commensals can also promote resistance to *E. coli*[6,8], Influenza virus[9], *K. pneumoniae*[10,11], *L. monocytogenes*[5], *S. aureus*[12], *S. pneumoniae*[10]. Despite these numerous studies detailed mechanistic understanding of the metabolic events that link commensal colonization with neutrophil functionality, remain incompletely understood.

Here, we show evidence that the proteomic signatures of neutrophils derived from mice with and without infection in the presence or absence of commensal colonization were significantly different.

[1]Department of Medicine, Division of Infectious Diseases, Mass General Brigham, Harvard Medical School, Boston, MA 02115, USA. [2]Department of Chemistry, University of Minnesota, Minneapolis, MN 55455, USA. [3]Center for Regenerative Medicine, Massachusetts General Hospital, Boston, MA 02114, USA. [4]Molecular and Cellular Biology Department, University of Guelph, Guelph, ON N1G 2W1, Canada. [5]Department of Obstetrics, Gynecology, and Reproductive Biology, Brigham and Women's Hospital, Harvard Medical School, Boston, MA 02115, USA. [6]Harvard University, Faculty of Arts and Sciences, Cambridge, MA 02138, USA. ✉e-mail: mgadjeva@fas.harvard.edu

Among the identified network of infection and colonization-induced proteins was a previously uncharacterized protein termed Prenylcysteine oxidase-1-like protein (Pcyox1l). We show that *Pcyox1l* deficiency inhibits the mevalonate pathway leading to reduced protein prenylation and cellular viability under homeostatic conditions. Consistently, *Pcyox1l* KO mice show elevated susceptibility to challenge with the gram-negative pathogen, *P. aeruginosa*. Cumulatively, these data reveal a commensal-dependent metabolic modulator of neutrophil function.

## Results

### Neutrophil proteomes are altered by microbiota and by infection

To generate a data base of neutrophil proteomes, quantitative proteomic analysis was performed on neutrophils derived from uninfected and *P. aeruginosa*–infected SW mice under both SPF and GF conditions. Quadruplicate samples led to the identification of 5588 proteins and filtering for valid values resulted in 3633 proteins used for further analysis. Biological replicate reproducibility ranged from 90% to 95%, demonstrating consistent protein recovery. 371 proteins showed significant differences in abundance upon comparison of PMN proteomes from non-infected SPF SW and GF SW mice (Fig. 1a). 837 proteins showed significant differences in abundance upon comparison of PMN proteomes derived from infected SPF SW and GF SW mice. A Principal Component Analysis (PCA) demonstrated clear clustering of the biological replicates (Fig. 1b and Supplementary Fig. 1) and distinct patterns of protein responses among microbiome-sufficient and deficient mice and treatments. Specifically, the largest separating component (component 1 at 32.7%) distinguished samples by colonization (SPF vs. GF) with the second component (component 2 at 10%) separating samples by infection status. These results illustrate dramatic

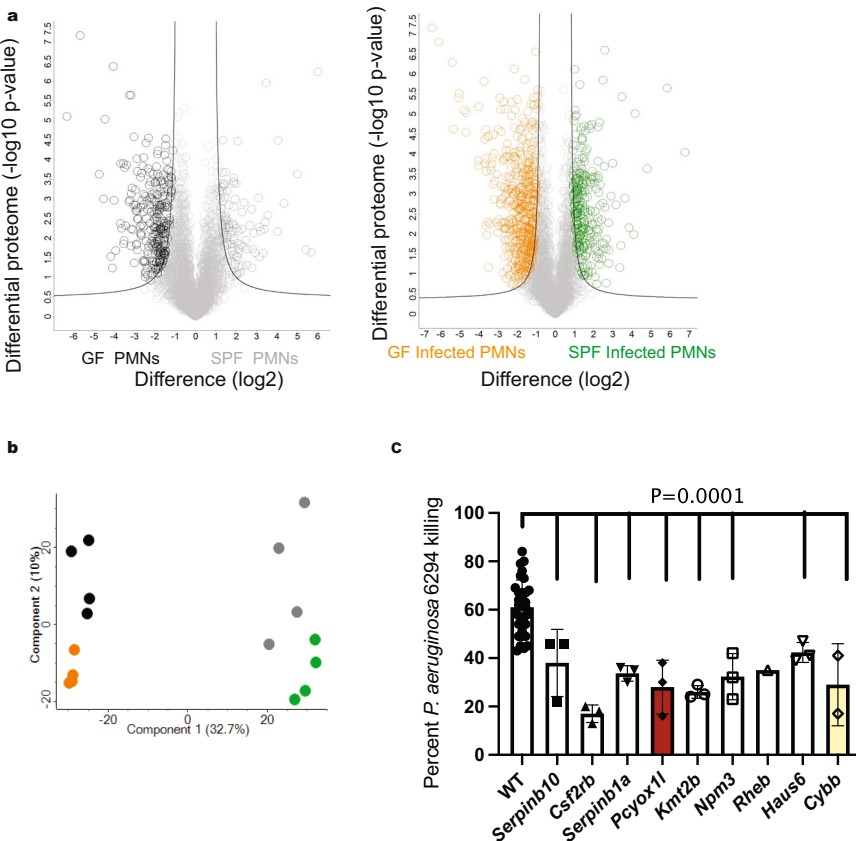

**Fig. 1 | Microbiota and infection alter neutrophil proteomes.** Neutrophils were purified from non-infected and *P. aeruginosa*-infected SPF and GF SW mice, lysates were trypsin digested, and proteomes solved using LC-MS/MS as in[4,55]. **a** Volcano plots displaying the differential protein abundance profiles. Two-tailed Student's *t*-test was performed to identify proteins with significant differential abundance ($p < 0.05$) between PMNs derived from non-infected and infected SPF SW and GF SW mice employing 5% permutation-based FDR filter. The left volcano plot presents the comparisons between PMNs from uninfected GF (black circles) and SPF SW (grey circles) mice (left panel) and the right plot represents a comparison between PMNs harvested from infected GF (orange circles) and SPF SW mice (green circles) (right panel). Significantly different proteins are highlighted in color. The x-axis represents fold difference ($\log_2$); the y-axis represents $-\log_{10}$ *p*-values. The experiment was done in quadruplicates. **b** Principle Component Analysis was performed to determine proteome differences at the experimental level. The experiment was done in quadruplicate. Each circle represents an independent biological sample. The black circles indicate PMNs harvested from GF mice, whereas the gray circles indicate PMNs harvested from SPF SW mice at baseline. The orange and green circles indicate PMNs harvested from infected GF and SPF SW mice, respectively.

(x-axis component 1 = 32.7%, y-axis component 2 = 10%). **c** Opsonophagocytic assays with *P. aeruginosa 6294* were carried out with matured WT neutrophils (open bar, *N* = 31), WT neutrophils CRISPRed for *SerpinB10* (grey bar, *N* = 3), *SerpinB1a* (grey bar, *N* = 3), *Csfr2b* (grey bar, *N* = 3), *Pcyox1l* (red bar, *N* = 3), *Npm3* (grey bar, *N* = 3), *Rheb* (grey bar, *N* = 1), *Haus6* (grey bar, *N* = 3), *Kmt2b* (grey bar, *N* = 3) where N depicts the number of successfully generated CRISPRed clones per target. Data are presented cumulatively from at least three experiments. Each individual data point represents an average value derived from two or three biological replicas per clone. Data for the mature WT and *Rheb* CRISPRed PMNs are derived from a single clone and represent biological replicas. The *Cybb* CRISPRed cells (yellow bar, *N* = 2) were used as a positive control. Data are presented as mean values with SD. Ordinary one-way ANOVA with an overall *P* = 0.0001 with Dunnett's multiple comparisons adjusted P values, α = 0.05. The nine individual comparisons are as follows: WT vs *SerpinB10* KO *P* = 0.0069, WT vs *Csfr2b* KO *P* < 0.0001, WT vs *SerpinB1a* KO *P* = 0.0009, WT vs *Pcyox1l* KO *P* < 0.0001, WT vs *Kmt2b* KO *P* < 0.0001, WT vs *Npm3* KO *P* = 0.0004, WT vs *Rheb* KO *P* = ns, WT vs *Haus6* KO *P* = 0.0459, WT vs *Cybb* KO *P* = 0.0013. Source data are provided as a Source Data file.

remodeling of the neutrophil proteomes in infected mice depending on commensal presence.

We selected 10 differentially upregulated proteins to explore their functional significance (Supplementary Table 1). We utilized a previously used by us CRISPR/Cas9 approach to generate three distinct neutrophil progenitor ER-Hoxb8 KO cell lines for each target. The ER-Hoxb8 progenitors can be matured in vitro to terminally-differentiated and functional neutrophils with characteristics comparable to primary, bone marrow-derived neutrophils including the normal morphology with multilobed nuclei and opsonophagocytic vacuoles[3]. Eight targets were successfully knocked out, out of those seven were targeted by all three guides and one target (e.g., Rheb) was targeted by a single guide (Suppl. Table 2). PMNs generated from the wild type (WT) and CRISPRed-out progenitor lines were examined for bactericidal properties. The loss of all targeted proteins resulted in neutrophils with significant reductions in opsonophagocytosis against *P. aeruginosa* (Fig. 1c). Cumulatively, these data pointed towards intracellular pathways controlling bactericidal functions of neutrophils.

## Pcyox1l deficiency decreases neutrophil bactericidal activities

We focused on an analysis of Pcyox1l-controlled biological pathways. This choice was based on the high degree of evolutionary conservation of Pcyox1l across vertebrates implicating functional importance. Both, mouse and human Pcyox1l proteins share 94% amino acid sequence identity and similar protein folding (Figs. 2a and 2b). Consistent with the homology data, the antibody reactive against the mouse Pcyox1l protein was also crossreactive with the human Pcyox1l (Fig. 2c). The *Pcyox1l* CRISPRed cell lines were validated via WB analysis under three different maturation conditions: vehicle, GCSF, and GCSF/GM-CSF (Fig. 2d, Supplementary Fig. 2a and b). When matured with the different stimuli in vitro, the WT and *Pcyox1l* CRISPRed cell lines generated banded and multilobed PMNs (Supplementary Fig. 3).

To examine the impact of Pcyox1l deficiency on neutrophil functions further, opsonophagocytic assays were carried out[13]. We used two different WT control neutrophil cell lines. One of them was engineered to express GFP and the other one was *GFP* knock-out CRISPR/Cas9 control (WT, H8) (Fig. 2e, left panel, compare the green bar versus the open bar). All three *Pcyox1l* KO clones showed decreased *P. aeruginosa 6294* killing irrespective of the guide used to generate the lines. Of note, the opsonophagocytic capacity of the in vitro matured control WT neutrophils was similar to those of primary bone marrow (BM) derived PMNs (Fig. 2e, middle and right panels, compare grey bars to open bars), an observation consistent with our previous reports validating the use of the ER-Hoxb8 neutrophil cell lines[3]. The diminished opsonophagocytic activities of *Pcyox1l* CRISPRed PMNs against *P. aeruginosa* correlated with reduced ROS release in the presence or absence of anti-*P. aeruginosa* MoAb (Supplementary Fig. 4A and B). Additional OPK experiments with *P. aeruginosa PAO1* showed a comparable decrease in the bactericidal potency of *Pcyox1l* CRISPRed PMNs ruling-out bacterial strain-specific phenotypes (Supplementary Fig. 4C). Cumulatively, *Pcyox1l* deficiency was associated with decreases in bactericidal properties of neutrophils.

## Pcyox1l deficiency results in reduced de novo protein prenylation

Protein prenylation is the post-translational transfer of a farnesyl or a geranylgeranyl moiety to one or two cysteine residues present near the C-terminus of a target protein. There are three types of prenylation; farnesylation catalyzed by farnesyltransferase (FTase), and type I and II geranylgeranylation catalyzed by geranylgeranyl transferase I and II (GGTase I and II) respectively[14]. Recently, GGTase III, has been identified with a very limited substrate scope[15–17]. Prenylated proteins, many of which are small GTPases, are crucial for cellar function and regulation. To examine levels of protein prenylation, *Pcyox1l* deficient and WT progenitor and GCSF-matured neutrophils were metabolically

labelled with C15AlkOPP[18,19], an analogue of the endogenous isoprenoid substrates with an embedded alkyne (Fig. 3a)[18–20]. This labeling method allows for the selective modification of prenylated proteins with an alkyne functionality that provides a handle for ligation with biotin-azide, which in turn allows for the selective enrichment of prenylated proteins though avidin pull down. These enriched proteins were then subjected to a bottom-up proteomic analysis for the identification and quantification of prenylated proteins, as well as for analyzing overall trends in prenylation levels. An overall increase in prenylation during the maturation of the neutrophil cells was observed with 44 prenylated proteins found in the GCSF-matured cells compared to 38 proteins found in the progenitor samples. Most of the proteins found were geranylgeranylated (Fig. 3b and c). 31 out of the total 38 identified proteins were significantly enriched in WT progenitors when compared to *Pcyox1l* KO progenitors. This pattern was also observed in the comparison of WT-GCSF-matured cells to *Pcyox1l*-KO-GCSF matured cells, with 39 out of 44 proteins enriched in the mature WT. The substantial decrease in prenylation levels in both, the *Pcyox1l* KO progenitors and *Pcyox1l* KO-GCSF-matured neutrophils, is further illustrated by the fact that no prenylated proteins were enriched in the *Pcyox1l* KO genotype.

String-based analysis of the relative to levels in the WT differentially prenylated proteins (Supplementary Table 3A and B) revealed a highly enriched in interactions network (Supplementary Fig. 5) ($p < 1.0e-16$). In the GCSF-matured WT cells the differentially upregulated prenylated proteins were involved in biological processes associated with respiratory burst (FDR $p = 0.02$), phagolysosome maturation (FDR $p = 0.0013$), phagolysosome assembly (FDR $p = 0.02$) and phagocytosis (FDR $p = 0.002$), suggesting a mechanistic connection between protein prenylation and cellular functionality.

To complement the metabolic labeling experiments, we performed metabolomic analysis to quantify free prenylpyrophosphate and prenylcysteine metabolites (Figs. 3d and 4). Prenylpyrophosphates are added onto proteins to generate their prenylated form, while prenylcysteine metabolites arise from the degradation of the prenylated proteins (Fig. 4a). In vitro vehicle-matured WT and *Pcyox1l* CRISPRed PMNs had relatively low levels of farnesylpyrophosphate (FPP) and significantly decreased gernaylgeranylpyrophosphate (GGPP) (Fig. 4b). Contrary to our expectations, a trend towards lower levels of geranylgeranylcysteine and farnesylcyteine metabolites was detected in vehicle-matured *Pcyox1l* CRISPRed PMNs when compared to WT PMNs. Significantly less free farnesylcysteine and geranylgeranylcysteine metabolites were present in the GCSF-matured *Pcyox1l* CRISPRed PMNs (Fig. 4c). The analysis of primary, bone marrow-derived PMNs showed similar decreases in geranylgeranylcysteine metabolites as the cell lines under steady state (Fig. 4d). Given that this decrease was observed despite the presence of the established prenylcystinelyase Pcyox1 in the *Pcyox1l* KO BM PMNs (Supplementary Fig. 6), we concluded that Pcyox1l is a key enzyme in the mevalonate pathway, with a more impactful role than the previously characterized enzyme Pcyox1. Cumulatively, data point to reductions in prenylcysteines in the *Pcyox1l* deficient PMNs under homeostatic conditions, indicative of significant reductions in the mevalonate pathway.

During infection the trend towards lower geranylgeranylcysteine metabolite levels was reversed. The geranylgeranylcysteine metabolites were significantly higher in the *Pcyox1l* KO PMNs derived from infected mice consistent with the expectations for Pcyox 1l catabolic activity (Fig. 4d).

A consequence of the altered prenylation pathway in the absence of Pcyox1l is a significant reduction in neutrophil viability. We observed a dramatic decrease in the longevity of *Pcyox1l* deficient PMNs illustrated by a 100-fold decrease in the frequencies of the recoverable viable PMNs (Supplementary Figs. 7 and 8). Pharmacologic blockade of the final step of maturation of prenylated proteins via cysmethynil mimicked *Pcyox1l* deficiency outcomes (Supplementary

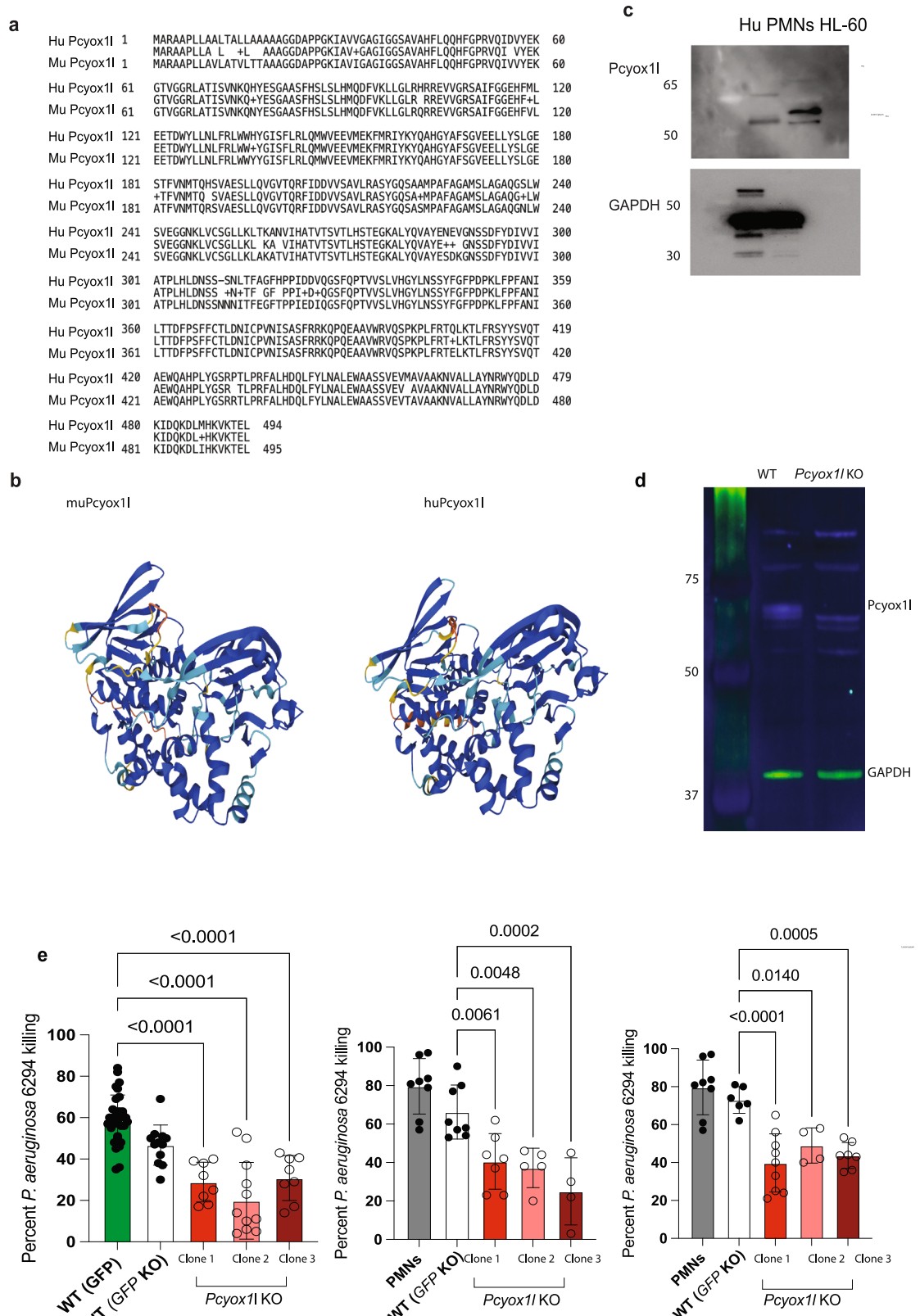

Fig. 8B), connecting blockade in prenylation to cellular outcomes. Notably, the viability of their in vitro-matured *Pcyox1l* CRISPRed PMNs was also significantly decreased despite treatments that typically extend neutrophil viability such as GM-CSF or GCSF (Supplementary Fig. 8C). Interestingly, CRISPRing out another target identified by our initial neutrophil proteome profiling experiment—*SerpinB1a*—showed a similar phenotype to that of *Pcyox1l* CRISPRed PMNs (Supplementary

Fig. 8C). The *Pcyox1l* CRISPRed PMNs also presented with diminished protein levels for *SerpinB1a*, confirming a link between the pathways (Supplementary Fig. 8D). The viability defect correlated with changes in the autophagy flux and autophagolysosomal degradation as evident by diminished LC3II and p62 levels in *Pcyox1l* CRISPRed PMNs irrespective of chloroquine treatments (Supplementary Figs. 9 and 10). Cumulatively, our data connect *Pcyox1l* deficiency to changes in

**Fig. 2 | Pcyox1l protein is evolutionary conserved and controls neutrophil bactericidal activities. a** Blast analysis of human Pcyox1l (Q8NBM8) and murine Pcyox1l (Q8C7K6) polypeptide sequence reveals 94% sequence identity. **b** Predicted structures of murine Pcyox1l and human Pcyox1l generated by Alpha-Fold. **c** WB analysis for Pcyox1l in human primary neutrophils and differentiated into granulocytes HL-60 cells. Data shows presence of Pcyox1l in human cells. Data are representative of two different experiments. **d** Representative WB analysis for Pcyox1l in WT and one of the generated *Pcyox1l* CRISPR-ed murine clones where the Pcyox1l expression was knocked out. Data validate CRISPRed *Pcyox1l* in the neutrophil cell lines. Data are respresentative of two different experiments. **e** OPK assays with *P. aeruginosa 6294* were carried out with BM-derived PMNs (grey bars), WT neutrophils expressing GFP (green bar), WT neutrophils CRISPRed for *GFP* (open bars), different *Pcyox1l* CRISPRed clones under 3 different maturation conditions including vehicle (red bars), 20 ng/ml GCSF (pink bars), and 20 ng/ml GCSF with 5 ng/ml GM-CSF (dark red bars) at MOI 0.01. Data are cumulative from at least five different experiments and plotted as mean values +/− SD. The individual symbols represent biological replicas. Left panel depicts OPK with vehicle-matured PMNs: GFP-expressing PMNs (*N* = 32), GFP CRISPRed PMNs (*N* = 12), *Pcyox1l*

CRISPRed clone 1 (*N* = 8), *Pcyox1l* CRISPRed clone 2 (*N* = 12), *Pcyox1l* CRISPRed clone 3 (*N* = 8). Ordinary one-way ANOVA with an overall *P* < 0.0001 followed by Dunnett's multiple comparison test: WT vs Clone 1 *P* < 0.0001, WT vs Clone 2 *P* < 0.0001, WT vs Clone 3 *P* < 0.0001. Middle panel depicts OPK data with BM-derived PMNs (grey bar, *N* = 8), GCSF-matured GFP CRISPRed PMNs (WT, *N* = 8), *Pcyox1l* CRISPRed clone 1 (*N* = 7), Pcyox1l CRISPRed clone 2 (*N* = 5), *Pcyox1l* CRISPRed clone 3 (*N* = 4). Ordinary one-way ANOVA with overall *P* = 0.0001 followed by Dunnett's multiple comparisons test: WT vs Clone 1 *P* = 0.006, WT vs Clone 2 *P* = 0.0048, WT vs Clone 3 *P* = 0.0002. Right panel depicts OPK data with BM-derived PMNs (grey bars, *N* = 8), GCSF and GM-CSF-matured GFP CRISPRed PMNs (WT, *N* = 6), *Pcyox1l* CRISPRed clone 1 (*N* = 9), *Pcyox1l* CRISPRed clone 2 (*N* = 4), *Pcyox1l* CRISPRed clone 3 (*N* = 7). Ordinary one-way ANOVA with an overall *p* = 0.0001 followed by Dunnett's multiple comparison test. WT vs Clone 1 *P* < 0.0001, WT vs Clone 2 *P* = 0.01, WT vs Clone 3 *P* = 0.0005. Source data are provided as a Source Data file. Cumulatively, data show that Pcyox1l is an evolutionary conserved protein with significant functions promoting opsonophagocytic activities of neutrophils.

prenylation, autophagy, and reduced viability under homeostatic conditions.

### *Pcyox1l* KO mice are more susceptible to keratitis

Given the dramatic phenotype of *Pcyox1l* KO neutrophils in vitro, we obtained *Pcyox1l* KO mice. The mice were characterized by complete blood counts as well as by flow cytometry to look at myeloid cells and neutrophils in the peripheral blood, spleen, and bone marrow. Under homeostatic (uninfected) conditions, the *Pcyox1l* KO mice did not show differences in white blood cell counts, absolute neutrophil counts, or hematocrit (Supplementary Figs. 11 and 12). The frequency of mature myeloid cells (CD11b[+], GR1[+]) by flow cytometry did not differ between WT and KO mice in the peripheral blood, spleen, or bone marrow (Supplementary Fig. 12).

We next tested the susceptibility to *P. aeruginosa*-induced ocular infection in vivo. Infected *Pcyox1l* KO mice showed significant ocular pathology exemplified by increased corneal opacity and hemorrhaging (Fig. 5a). Haematoxylin and eosin staining of corneal sections indicated sloughing of the corneal epithelial layer at the site of infection, increased neutrophil-dominant infiltration, and hemorrhaging, consistent with worse disease (Fig. 5b). The *Pcyox1l* KO mice had a significantly higher corneal bacterial burden when compared to WT littermates infected with MPAO1 (Fig. 5c). To rule out bacterial strain-specific effects, infection experiments were repeated with the clinical isolate 6294 (Fig. 5d) which showed an overall worse disease with milder but consistent differences between the infected *Pcyox1l* KO and WT mice.

Analysis of peripheral blood in the infected mice showed significant reduction in WBC and PMNs levels but no changes in hematocrit in the infected *Pcyox1l* KO mice (Fig. 6a). To examine PMN recruitment to the infected site, frequencies of infiltrating PMNs were analyzed by flow cytometry. The infected *Pcyox1l* KO mice showed increased frequencies of viable PMNs (CD45[+]CD11b[+]Ly6C[+]Ly6G[+]) in the infected corneas (Fig. 6b and c), consistent with the morphological analysis. To determine if this increase was due to alterations in neutrophil-recruiting cytokines in the infected tissues, the relative cytokine presence was quantified. There were no major differences in IL-6, IFN-γ, TNF-α, KC, or MIP-2 levels, and mild elevation of GM-CSF levels (Fig. 6d).

Adoptive transfer experiments were carried out where CD18KO mice were injected with WT or *Pcyox1l* CRISPRed PMNs, infected with *P. aeruginosa* 6294, bacterial burden, and pathology measured at 24 h post-challenge. The CD18 KO mice that received Pcyox1l CRISPRed PMNs showed significantly higher bacterial presence and worse disease-associated pathology when compared to the CD18 KO mice receiving WT PMNs (Fig. 6e) indicating that the *Pcyox1l* deficient PMNs

had compromised functionality. Cumulatively these data reveal a defect in anti-bacterial innate immune responses in the absence of Pcyox1l exemplified by higher bacterial burden in the infected tissues despite the increased infiltrating PMNs.

## Discussion

Protein prenylation typically promotes an association of the prenylated proteins with plasma and endomembranes where they interact with other protein partners[21]. While it is relatively well-known how prenyl moities are loaded onto proteins, the catabolic pathway responsible for the degradation of the prenylated proteins is incompletely understood. The last step of the catabolic processing of the prenylated proteins is the degradation of prenylcysteine metabolites to farnesal or geranylgeranial, hydrogen peroxide, and cysteine. This step is performed by an enzyme termed prenylcysteine oxidase (Pcyox 1)[22,23]. Up-till-today there are no other studied enzymes with a similar function to Pcyox1. Here, we offer experimental evidence for the discovery of another enzyme, namely, Pcyox1l. We find that the loss of Pcyox1l affects the mevalonate pathway leading to changes in protein prenylation and ultimately affects neutrophil bactericidal activities. In order to reach these conclusions, we generated reagents including *Pcyox1l* KO mice and *Pcyox1l* CRISPRed neutrophil progenitor cell line models.

We find that under homeostatic conditions, the *Pcyox1l* deficiency is associated with reductions in de novo prenylation of a network of small GTPases, reductions in prenylpyrophosphate, and prenylcysteine metabolites in in vitro matured *Pcyox1l* CRISPRed and *Pcyox1l* KO BM-derived PMN. These metabolic characteristics of *Pcyox1l* KO PMNs correlate with reduced viability and alterations in autophagy. In contrast, during infection the BM-derived *Pcyox1l* KO PMNs show inability to catabolize geranylgeranylcysteine when compared to WT BM PMNs harvested from *P. aeruginosa* infected mice, consistent with the expected enzymatic function for Pcyox1l. These metabolic changes are accompanied with reduced ROS release, diminished bactericidal activities in vitro, and elevated pathology. Cumulatively, our findings show that the *Pcyox1l* deficiency presents with altered prenylation and elevated susceptibility to infection (Fig. 7).

The homeostatic phenotype of the *Pcyox1l* KO PMNs is puzzling. We expected to observe a build-up of subtrates as was the case for the *Pcyox1* deficiency which showed increased isoprenoid metabolites at steady state[24]. This prompted the question why the loss of a catabolic enzyme, namely Pcyox1l, did not follow the same phenotype as the loss of Pcyox1 but instead showed an overall reduction of the prenylation pathway. Is it possible that the phenotype is an indirect effect of Pcyox1l loss? When analyzing

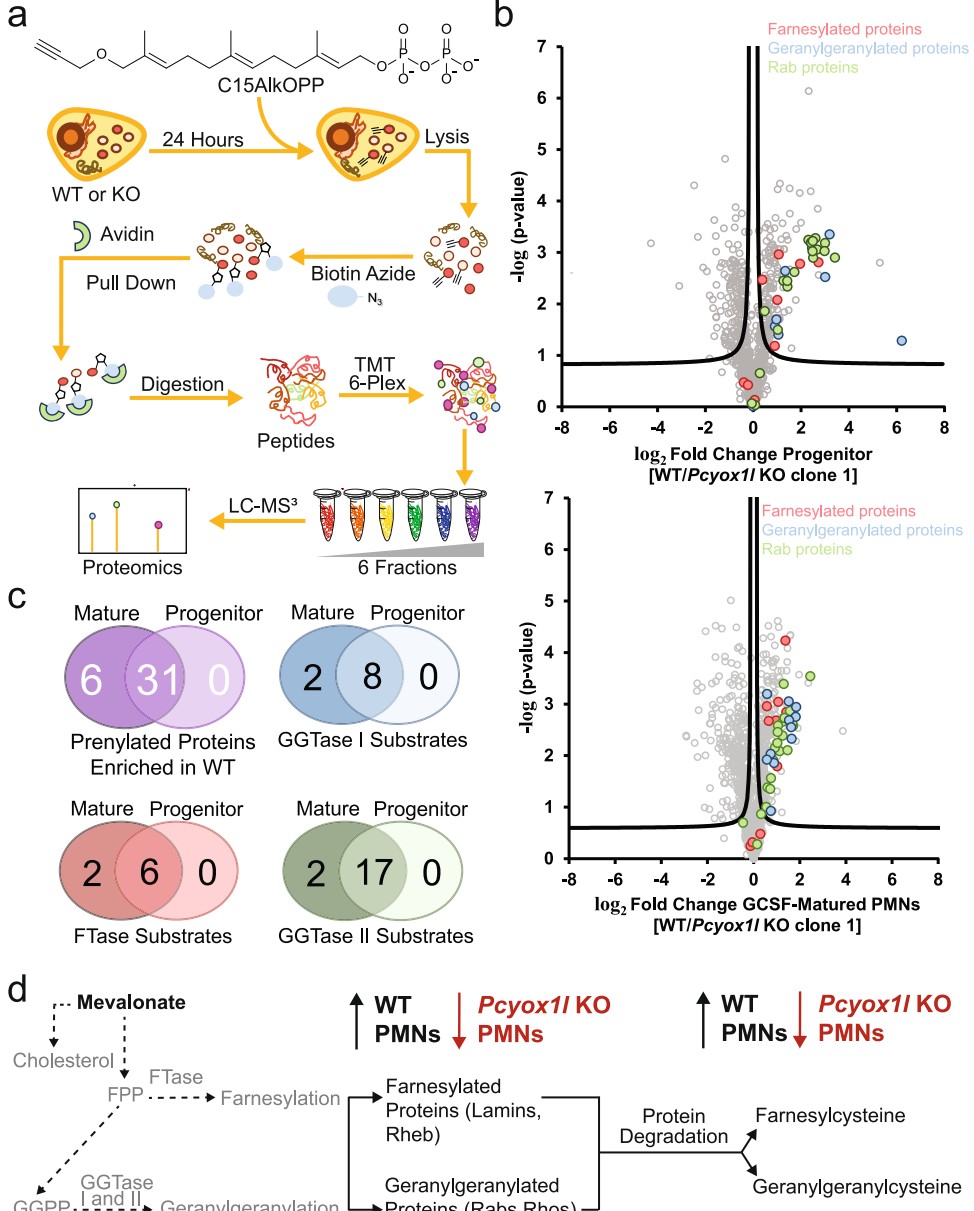

**Fig. 3 | Pcyox1l deficiency affects neutrophil prenylomes. a** Metabolic labeling experimental workflow. WT and *Pcyox1l* CRISPRed progenitors and matured with GCSF neutrophils were fed with CC15AlkOPP probe o/n to allow incorporation of the probe in de novo formed prenylated proteins. Cells were lyzed and the prenylated proteins captured via click-chemistry mediated biotinylation and pulldown. Samples are multiplexed and prenylomes solved on LC-MS. **b** Volcano plots showing differential prenylated proteins in WT (H8) and *Pcyox1l* CRISPRed GMPs (top) and GCSF-mature neutrophils (bottom) in vitro. FDR = 5%, s0 = 1. Farnesylated proteins (red), type 1 geranylgeranylated proteins (blue), and type II geranylgeranylated proteins (green), non-prenylated proteins found in dataset (grey). Prenylated proteins were found enriched in the WT (right of volcano). This suggests that prenylation is heavily down regulated when *Pcyox1l* is deficient. Overall, 44 known prenylated proteins were found in the GCSF-mature dataset and 38

prenylated proteins were found in the GMP dataset. **c** Analysis of the impact of neutrophil maturation on cellular prenylation. Three Venn diagrams depicting significantly different prenylated proteins distributions in the total dataset. Overall, there were more proteins found in the GCSF-mature cell compared to the GMP (puple). FTase substrates (red), GGTase I protein substrates (blue), and GGTase II protein substrates (green). **d** Diagram depicting the prenylation pathway. Farnesylpyrophosphate and geranylgeranylpyrophosphate are synthesized via the mevalonate pathway, which are then transferred onto proteins via farnesyltransferase (FTase) or geranylgeranyl transferases (GTase I, II, and III). Prenylated proteins are degraded releasing prenylcysteines (farnesylcysteine and geranylgeranylcysteine). Cumulatively, data indicate significant impairment of the de novo prenylation in the absence of Pcyox1l.

autophagy in WT and *Pcyox1l* CRISPRed out PMNs, we noted significant reductions in the autophagy flux in the *Pcyox 1l* CRISPRed PMNs and decreases in the relative levels of the signaling scaffolding protein p62/SQSTM1, a selective receptor for mitophagy. Because p62 deficiency is connected to mitochondrial dysfunction[25] which causes an overall reduction of the mevalonate pathway, we entertain the idea that the *Pcyox1l* deficiency through alterations in autophagy/mitophagy diminish global prenylation[26].

The connection between the Pcyox1l enzyme deficiency and alterations in autophagy/mitophagy is further reinforced by our data on the state of Rheb prenylation, another protein identified by the initial proteome screening experiment. Rheb is a farnesylated protein that controls mTORC1 activation[27,28] and we found that the *Pcyox1l* deficiency resulted in reductions of Rheb farnesylation. Rheb controls mitochondrial renewal and prevents accumulation of damaged mitochondria[29]. Transient overexpression of Rheb causes reduction in

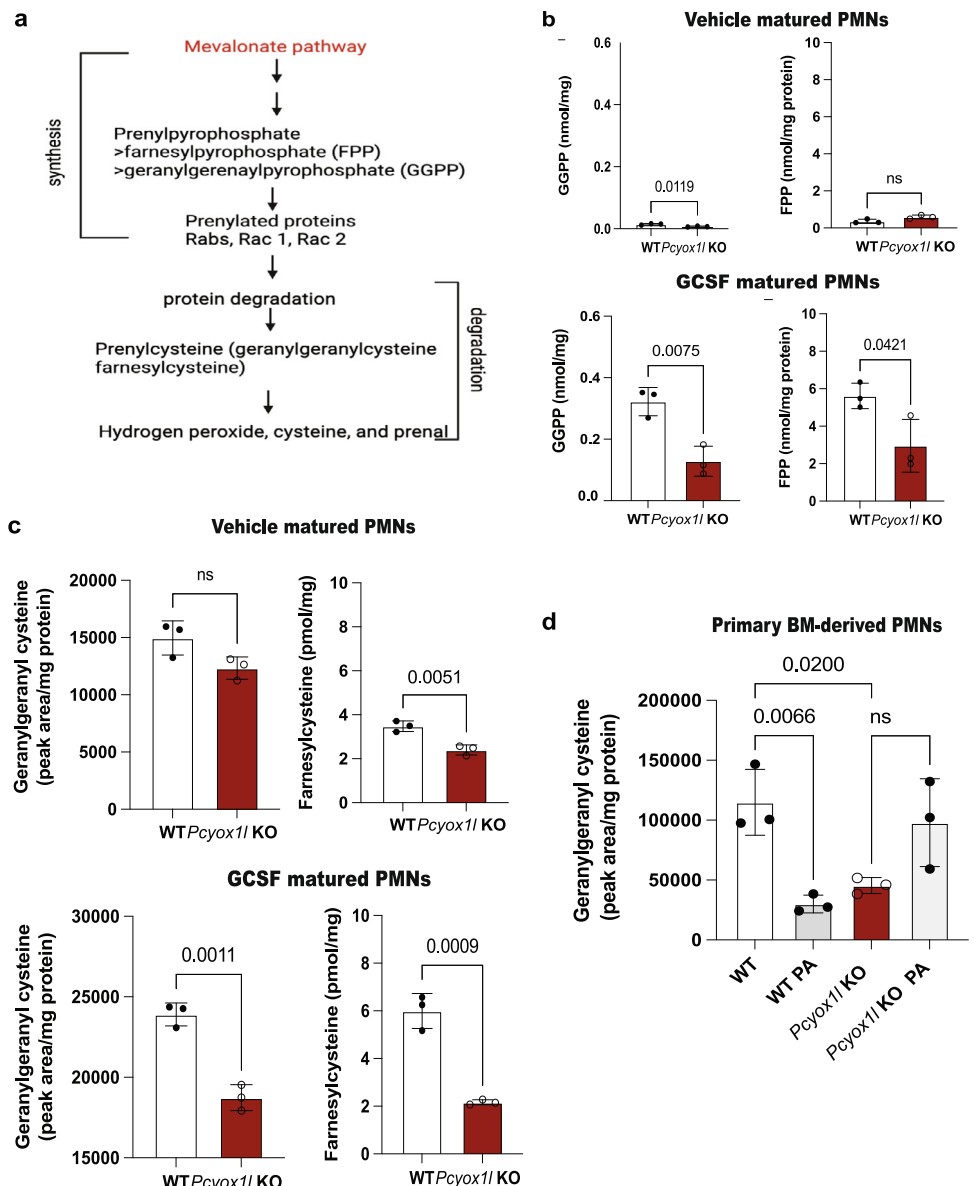

**Fig. 4 | *Pcyox1l* deficient neutrophils show significant alterations in prenylpyrophosphates (FPP and GGPP) and prenylcysteine metabolites at steady state and during infection. a** Schematic diagram of the mevalonate pathway leading to the synthesis of prenylated proteins and their free metabolites. **b** Metabolic quantification of FPP and GGPP in lysates from vehicle and GCSF-matured WT and *Pcyox1l* CRISPRed PMNs. $12 \times 10^6$ neutrophil progenitors were matured with 20 ng/ml GCSF. Mature neutrophils were harvested 3 days after initiating the maturation schedule, lysates processed for relative FPP and GGPP presence. Data are plotted as mean and SD. Each symbol represents a biological replica. FPP measurements in vehicle matured WT PMNs ($N = 3$) and *Pcyox1l* CRISPRed PMNs ($N = 3$). Unpaired two-tailed *t*-test, $P = $ ns. GGPP comparisons in vehicle−WT ($N = 3$) and *Pcyox1l* CRISPRed PMNs ($N = 3$). Unpaired two-tailed *t*-test, $P = 0.01$. FPP measurements in GCSF−matured WT PMNs ($N = 3$) and *Pcyox1l* CRISPRed PMNs (N = 3). Unpaired two-tailed *t*-test, $P = 0.04$. GGPP comparisons in GCSF-matured WT ($N = 3$) and *Pcyox1l* CRISPRed PMNs ($N = 3$). Unpaired two-tailed *t*-test, $P = 0.007$. **c** Metabolic quantification of free prenylcysteine (farnesylcysteine and geranylgeranylcysteine) metabolites in lysates from vehicle- and GCSF-matured WT (open bars) and Pcyox1l CRISPRed PMNs (red bars). Each symbol represent a biological replica. Data are plotted as mean and SD.

Geranylgeranylcysteine comparisons in vehicle-matured WT ($N = 3$) and *Pcyox1l* CRISPRed PMNs ($N = 3$). Unpaired two-tailed *t*-test, $P = $ ns. Farnesylcysteine comparisons in vehicle-matured WT ($N = 3$) and *Pcyox1l* CRISPRed PMNs ($N = 3$). Unpaired two-tailed *t*-test, $P = 0.005$. Geranylgeranylcysteine comparisons in GCSF-matured WT ($N = 3$) and *Pcyox1l* CRISPRed PMNs ($N = 3$). Unpaired two-tailed *t*-test, $P = 0.001$. Farnesylcysteine comparisons in GCSF-matured WT ($N = 3$) and *Pcyox1l* CRISPRed PMNs ($N = 3$). Unpaired two-tailed *t*-test, $P = 0.0009$. **d** Metabolic quantification of free geranylgeranylcysteine metabolites in lysates from primary BM PMNs isolated from non-infected WT (open bar, $N = 3$), *Pcyox1l* KO (red bar, $N = 3$), infected WT (grey bar, $N = 3$), and infected *Pcyox1l* KO (light grey bar, $N = 3$) mice. Symbols indicate biological replicas. Data are plotted as mean +/− SD. Ordinary one-way ANOVA with an overall $P = 0.006$ followed by Sidak's multiple comparison test with adjusted $P = 0.02$ for the WT to *Pcyox1l* KO PMNs comparison and $P = 0.0066$ for the infected WT versus *Pcyox1l* KO comparison. Source data are provided as a Source Data file. Cumulatively, data indicate that the *Pcyox1l* deficiency has opposing effects in homeostasis and during infection. *Pcyox1l* KO PMNs show reductions in prenylcysteines at steady state conditions and inability to catabolize prenylcysteines during infection.

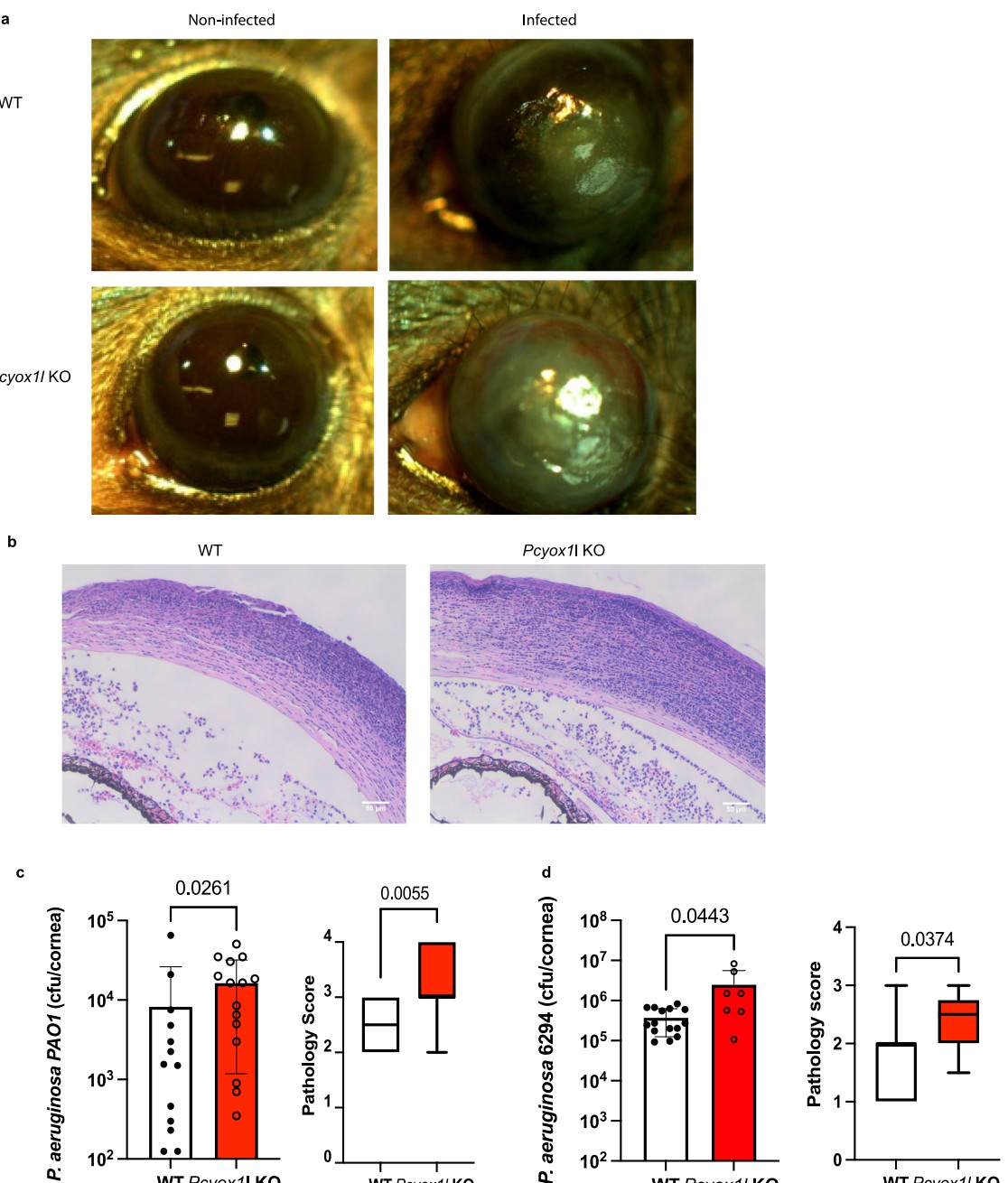

**Fig. 5 | *Pcyox1l* KO mice show elevated susceptibility to *P. aeruginosa*-induced keratitis. a** Representative images from non-infected and infected with *P. aeruginosa* 6294 WT and *Pcyox1l* KO mouse eyes out of two independent experiments. **b** Representative images of haematoxyllin and eosin stained-corneal sections from infected with *P. aeruginosa* 6294 WT and *Pcyox1l* KO mice. Sections were taken from infected corneas at 20 h post-challenge. Images were captured with Olympus at 40x. Scale bar 50 µm. Images are representative of two independent experiments where sections from infected WT (*N* = 6) and Pcyox1l KO mice (*N* = 7) were analyzed. **c** WT littermates (open bars) and *Pcyox1l* KO (red bars) mice were infected with 1 × 10⁷ CFU/eye *P. aeruginosa* PAO1 and bacterial burden quantified in the infected corneas at 20 h post-infection. Each symbol represents an individual animal. Unpaired two-tailed t-test with Welch's correction, *P* = 0.026. Pathology scores are

presented as boxes with whiskers. Boxes show 25% to 75% percentiles and whiskers show the minimum to maximum with the middle line reflecting the median. Mann–Whitney, *P* = 0.005. **d** WT littermates (open bars, *N* = 7) and *Pcyox1l* KO mice (red bars, *N* = 5) were infected with 1 × 10⁵ CFU/eye *P. aeruginosa* 6294 and bacterial burden quantified in the infected corneas at 20 h post-infection. Each symbol represents an individual animal. Mann–Whitney *p* = 0.04. Pathology scores. Mann–Whitney, *P* = 0.03. Boxes show 25% to 75% percentiles and whiskers show the minimum to maximum and the middle line reflecting the median. Mann–Whitney, *P* = 0.005. Data are cumulative from two experiments. Source data are provided as a Source Data file. Cumulatively, data show increased susceptibility to infection in the absence of *Pcyox1l*.

mitochondrial mass accompanied with LC3 deposition on mitochondrial membrane[29]. Cumulatively, we suggest that the overall decrease in the de novo prenylation of Rheb in *Pcyox1l* CRISPRed matured PMNs impaires autophagy/mitophagy reducing the flux through the mevalonate pathway under homeostatic conditions.

In contrast to the homeostatic state, infection induced degradation of geranylgeranylcysteine substrates in the WT BM-derived PMNs, but not in the *Pcyox1l* KO PMNs. These observations are consistent with the expectations for Pcyox1l enzymatic function and confirm the altered metabolism in the *Pcyox1l* deficient PMNs.

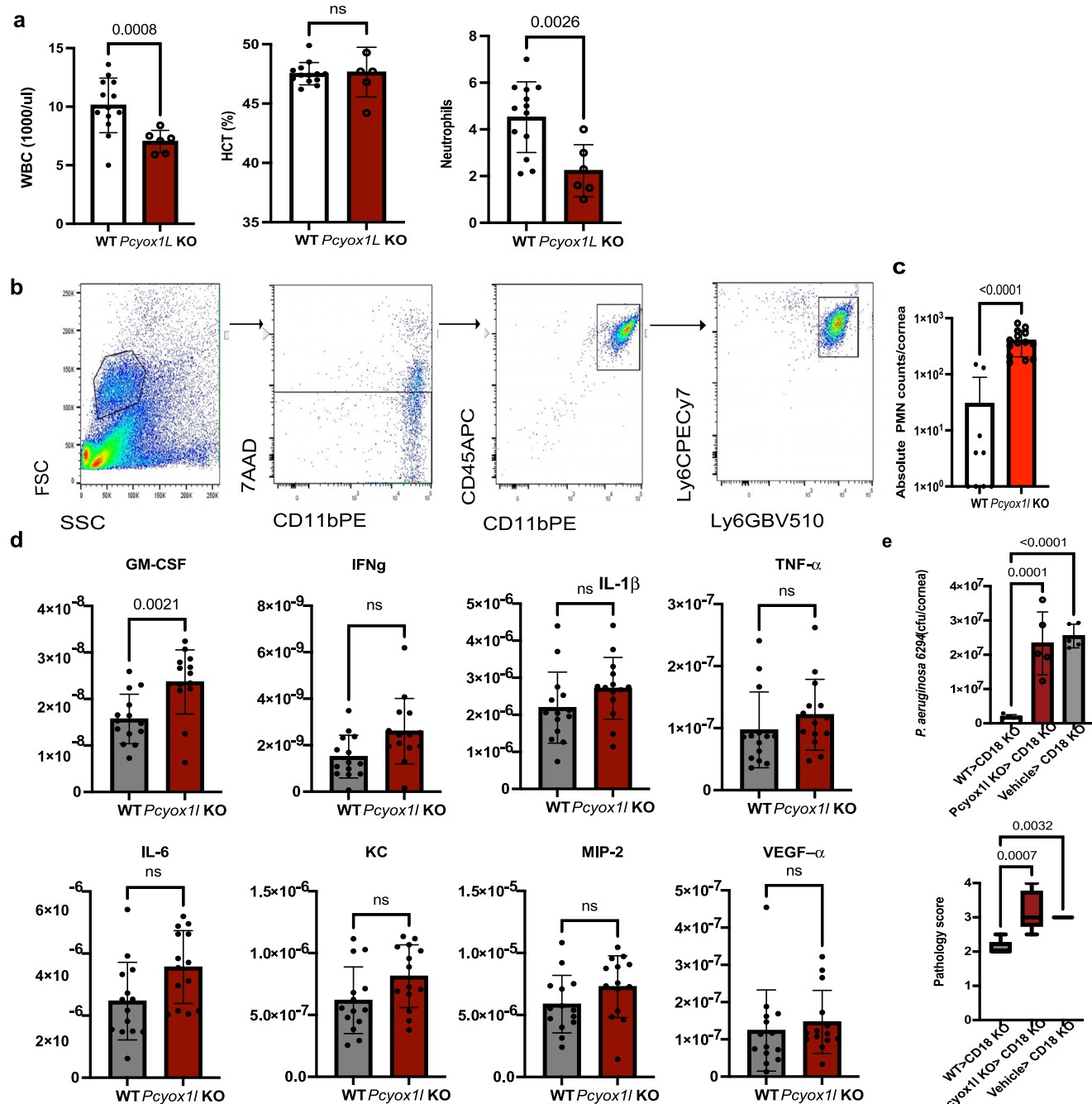

**Fig. 6 | *Pcyox1l* KO neutrophils show impaired bactericidal functions.**
**a** Hematopoietic characterization of infected *Pcyox1l* KO (red bar, *N* = 5) and WT (open bar, *N* = 13) littermates: white blood cell counts (WBC), hematocrit and neutrophils. Each symbol represents a biological replica. Unpaired two-tailed Student's *t*-test with Welch's correction. *P* = 0.008, *P* = ns, *P* = 0.002 Bonferonni's correction for multiple comparisons *P* = 0.01. **b** Flow cytometry analysis of infected corneas from WT (open bar, *N* = 10) and *Pcyox1l* KO (red bar, *N* = 17) mice. Samples were stained with CD45, Ly6C, Ly6G, 7AAD. Gating strategy and representative histograms are shown. **c** Absolute PMN counts CD45⁺7AAD⁻ Ly6C⁺Ly6G⁺ neutrophils were compared. Unpaired two-tailed *t*-test with Welch's correction, *P* = 0.0005. Each symbol represents a biological replica. **d** Corneal tissue lysates from infected WT (grey bar, *N* = 14) and *Pcyox1l* KO (red bar, *N* = 14) mice were quantified for GM-CSF (*P* = 0.002), IFN-γ, IL-1β, IL-6, KC, MIP-2, TNF-α, and VEGF-α by MSD. Unpaired two-tailed *t*-test with Welch's correction. Bonferonni correction

for multiple comparisons *P* = 0.006. Each symbol represents a biological replica. All data are cumulative from two experiments. **e** Adoptive transfer of *Pcyox1l CRISPRed* PMNs and WT PMNs into CD18KO mice. 1 × 10⁶ spontaneously in vitro matured *Pcyox1l* KO and WT PMNs were adoptively transferred to CD18 KO mice 6 h prior to bacterial challenge. Bacterial burden (**a**) and pathology (**b**) were quantified. Data are biological replicas. Ordinary one-way ANOVA with an overall *p* = 0.0001 followed by Dunnett's multiple comparison test. WT > CD18 KO (open bar, *N* = 5) vs Pcyox1l KO > CD18 KO (red bar, *N* = 5) *P* = 0.0001 and WT > CD18 KO vs Vehicle > CD18 KO (grey bar, *N* = 5) *P* = 0.0001. Pathology scores. Pathology scores are presented as boxes with whiskers. Boxes show 25% to 75% percentiles and whiskers show the minimum to maximum range with the middle line reflecting the median. Mann–Whitney, *p* = 0.0007. Data are from two experiments. Source data are provided as a Source Data file. Data highlight increased recruitment of neutrophils in the infected corneas, consistent with impaired function.

**Fig. 7 | Current working model.** Commensal presence drives expression of Pcyox1l in neutrophil progenitors and mature neutrophils via currently unknown mechanisms. The flux into the prenylation pathway is significantly decreased in the *Pcyox1l KO* PMNs under homeostatic conditions illustrated by reductions in free isoprenepyrophosphate metabolites (substrates for the generation of prenylated prteins), de novo prenylated small GTPases, and isoprenecysteine metabolites (products of the degradation of the prenylated proteins) (blue arrows). These alterations in the prenylation pathway lead to limited *Pcyox1l KO* PMN viability. During infection, the *Pcyox1l* KO PMNs show impaired catabolism of geranylger-anylcysteine substrates and reduced bactericidal activities. Consistently, the *Pcyox1l* KO mice show increased susceptibility to *P. aeruginosa*-induced infection due to diminished neutrophil functionality. Created with BioRender.com.

The reported metabolic changes are concurrent with changes in PMN functionality. A limitation of this work is our inability to directly examine the connection between the Pcyox1l enzymatic activity and PMN functionality. Performing Pcyox1l reconsitution experiments in PMNs is challenging as myeloid progenitor stem cell lines are notoriously difficult to transfect. The current technology is able to transfect only 5% of stem cells including the most recent approaches[30]. Insufficient numbers of mature neutrophils can be transfected and introduction of exogenous DNA into PMNs changes their responsiveness[31–33].

Despite these limitations, our data strongly suggest a link between Pcyox1l enzyme function and PMN's functionality. This connection is illustrated by our experiments where isoprenylcarboxymethyl trans-ferase (ICMT) enzyme activity is inhibited and cellular viability mon-itored. ICMT is an enzyme that catalyzes the last step of protein prenylation by transferring methyl groups onto prenylated cysteines. Inhibition of ICMT via cysmethynil reduces cellular viability in WT PMNs, a phenotype similar to the *Pcyox1l* CRISPRed PMNs. Published data point to reductions of Myd88 signaling and TLR-trigered imflammatory responses by inhibiton of ICMT[34].

Interestingly, silencing of *Pcyox1* in HepG2 cells reduces ROS release while overexpression of Pcyox1 promotes ROS release in CHO cells[35], displaying a phenotype similar to that of the *Pcyox1l* deficiency. Cumulatively, these data strengthen our arguments that the *Pcyox1l* deficiency through changes in prenylation affects PMN func-tion (Fig. 7).

Defects in PMN bactericidal activities are typically accompanied with elevated PMN recruitment to the infected corneas and our data shows consistent increase in PMN presence in the infected *Pcyox1l* KO

mice. Somewhat surprising is the finding that the neutrophil recruiting cytokines in the infected corneas were not dramatically elevated with the sole exception for GM-CSF. These data are intriguing as GM-CSF signaling extends PMN longevity and improves bactericidal functions. Consistently, we observe diminished PMN viability and opsonopha-gocytic killing upon GCSF/GM-CSF stimulations in vitro, illustrating that the *Pcyox1l* KO PMNs are unable to respond to GM-CSF. Cumula-tively, our data suggest that the *Pcyox1l* deficiency impairs PMN functions.

The majority of the presented work was focused on the analysis of the functional significance of Pcyox1l. However, our initial proteome screening experiment demonstrated a profound remodeling of PMN proteomes upon commensal colonization and infectious challenge, prompting several important questions: what is the impact of the other detected differentially present proteins on neutrophil functionality; and do the identified differentially expressed proteins function intra-dependently. Our data supports the concept that the commensal presence and infection upregulate proteins connected in a functional network. For example, Serpin B1a is a protein which appeared differ-entially expressed in PMNs upon commensal colonization and infec-tious challenge. Neutrophils from *SerpinBa1* KO mice have reduced viability that renders animals susceptible to *P. aeruginosa*-induced lung infection[36–39]. Excitingly, we found that SerpinB1a transcripts and protein levels were significantly diminished in GCSF or GCSF/GM-CSF matured *Pcyox1l* CRISPRed PMNs (Supplementary Fig. 8). Further, the decreased vability of *Pcyox1l C* RISPRed PMNs was comparable to that of *Serpin B1a* CRISPRed PMNs, illustrating a functional connection between Pcyox1l and SerpinB1a pathways. Cumulatively, the described network of commensal-induced proteome changes suggest events promoting PMN longevity and function[40].

It is tempting to speculate that our discovery of Pcyox1l relates to commensal-induced training of neutrophils[41,42]. The concept of trained immunity garnered significant experimental support recently and relates to changes in monocytic and neutrophil functionality. Trained immunity reflects metabolic reprogramming of mouse BM LT-HSCs which is maintained in mature monocytes and granulocytes through epigenetic mechanisms affecting the mevalonate pathway[43–45]. Phar-macologic blockade of the mevalonate pathway abrogates trained immunity in progenitors and mature myeloid cells. Myeloid training can be induced by gut colonization with *C. albicans* or upon challenge with β-glucan or zymozan[41,46]. Consistently, the lack of commensal gut colonization causes not only mild neutropenia but also important alterations in PMN functions. PMNs derived from GF mice have impaired killing of *P. aeruginosa, E.coli, and K.pneumoniae* when compared to BM derived PMNs from SPF mice[4,6]. It is tempting to propose that a key aspect of trained immunity is upregulation of Pcyox1l in the relevant cell types. To this end, we cannot exclude that the Pcyox1l deficiency affects functionally other cellular compart-ments in addition to neutrophils. This is a major focus for future studies.

The implications of our findings are far-reaching as they provide a framework for future research linking cellular metabolism to func-tional outcomes with a specific focus on prenylation. Historically, interest in studying prenylation stemmed from the finding that this modification was necessary to maintain malignant activity of oncogenic Ras proteins[47]. Today, prenylation is recognized as a main biochemical mechanism for altered cellular responses in neurode-generative diseases such as Alzheimer's[48], Parkinson's disease[49], Mul-tiple Sclerosis[50], mitochondrial diseases[51], osteoporosis[52], viral, and parasitic infections, therefore understanding the contribution of Pcyox1l in the maintenance of health is warranted. Cumulatively, we uncovered a global modulator of the prenylation pathway, which sig-nificantly impacts mature neutrophil functionalities thereby connect-ing commensal presence with prenylation metabolism and resistance to infection.

## Methods

### Mice

Mice were housed and bred at the MCP Animal Care Facility. Seven to nine week-old Specific Pathogen free (SPF) and Germ free (GF) Swiss Webster (SW) mice were obtained from the gnotobiotic core facility (BWH). The CD18 KO breeders were purchased from Jackson Labs (line 002128). The Pcyox1l mouse line 49020 was a product of CRISPR targeting to zygotes, developed by the Knockout Mouse Production and Phenotyping project (KOMP2) at the Jackson Laboratory, and deposited to the MMRRC. Pcyox1l KO mice (line 49020) were cryo-ecovered and intercrossed to generate Pcyox1l KO and WT littermates for experiments. The Pcyox1l KO are on C57Bl6/J background. Seven to nine-week old mice of both genders were used throughout the experiments. The individual number of mice per experiment are listed in the figure legends. All cohort predictions were based on Power G analysis to obtain significance level of $p < 0.05$ at 80% probability. Sex as a biological variable was not considered as prior experiments in mice did not point to sex as a determinant affecting clinical outcomes of P. aeruginosa-induced keratitis[53]. Similarly, there was no sex-bias reported for human disease severity[54].

### Genotyping

Genotyping was conducted using the following sequences: 49020-mutF2: 5′ AACACTTCCCTCACTCTTGGCTAAGC 3′ (mutant forward primer), 49020-comR2: 5′ CCTGCCACCCTCTTACTCGTCG 3′ (common reverse primer), and 49020-wtF: 5′ ACAGTTCAATGCAGGCTCAG 3′ (WT forward primer) (Integrated DNA Technologies (IDT), Coralville, IA, USA). The PCR reaction included Taq DNA Polymerase (Invitrogen, 18038-042, Waltham, MA USA), PCR reaction buffer (ThermoFisher, B65, Waltham, MA USA), 50 mM MgCl2 (ThermoFisher, AB0329, Waltham, MA, USA), 2.5 mM dNTPs, PCR grade water (10297-018, Invitrogen, Waltham, MA, USA). Mouse ear biopsies were digested in 1x of a 10x Digestion buffer containing 50 µL of 2-Mercaptoethanol (Sigma Aldrich, cat#M3148, St Louis MO,USA), 1.5 M Tris base (Sigma Aldrich, 77861, RO, St Louis MO,USA), 2 M (NH4)₂SO₃ (Sigma Aldrich, A4418, St Louis MO, USA), 1 M MgCl₂ (Sigma Aldrich, 1374248, St Louis MO, USA), and 100% Triton X-100 (Sigma Aldrich, T8787, St Louis MO,USA) for 3 min while heated in a 95 °C water bath. 20 mg/mL of Proteinase K (Qiagen, 158918, Hilden, Germany) was added immediately after digestion and samples were left in a 55 °C water bath for enzymatic digestion. Samples were reheated at 95 °C to arrest the digestion. PCR was run using the following steps: initiation at 94 °C for 2 min, denaturation at 94 °C for 10 s, annealing starting at 65 °C for 1 cycle and then decreasing the annealing temperature with 1 °C per cycle for 30 s, elongation at 68 °C for 2 min, denaturation at 94 °C for 15 seconds for 10 cycles, followed by 25 cycles of annealing for 55 °C for 30 s, elongation at 68 °C for 2 min, denaturation at 94 °C for 15 s. PCR bands were resolved on 2% Omnipur agarose (Sigma Aldrich, 2120, St Louis, MO USA) gels with Ethidium bromide 10 mg/ml (Invitrogen, 15585-011, Waltham, MA, USA), 1X TAE buffer (10X TAE buffer, Invitrogen, Ultra pure, 15558-026, Waltham, MA, USA) ran for 35 min at 90 volts.

### Bacterial strains and inocula

Invasive Pseudomonas aeruginosa clinical isolate 6294 and laboratory strain PAO1 were used throughout the experiments. The bacterial strains were grown overnight at 37 °C on trypticase soy agar plates (BD, BD 221261, Franklin Lakes, NJ, USA) supplemented with 5% sheep blood. All infectious suspensions were prepared in Hank's balanced salt solution (HBSS, HyClone Laboratories Inc, SH3058802, Logan, UT, USA) and used for assays.

### Infection model

Mice were anesthetized with intraperitoneal ketamine HCL (Zoetis, EA2489-564, Parsippany, NJ, USA) and xylazine injections (Lloyd Inc, Sc-362950Rx, Shenandoah, IA, USA). Corneas were scratched with 25 G

needel tip thrice generating three 5 mm scratches. An inoculum of $1 \times 10^7$ cfu of P. aeruginosa PAO1 and $1 \times 10^6$ cfu of P. aeruginosa 6294 was delivered in 5 µL onto the eye[4,55]. For evaluation of corneal pathology, daily scores were recorded by an observer unaware of the experimental conditions based on the following scoring system using a graded scale of 0 to 4 as follows: 0, eye macroscopically identical to the uninfected contra-lateral control eye; 1, faint opacity partially covering the pupil; 2, dense opacity covering the pupil; 3, dense opacity covering the entire anterior segment; and 4, perforation of the cornea, phthisis bulbi (shrinkage of the globe after inflammatory disease), or both. To quantify P. aeruginosa levels, corneas were suspended in DMEM/F12 with 0.05% Triton X100, serially diluted, and plated on P. aeruginosa selective McConkey agar plates (Thermo Fisher Scientific, R01552, Waltham, MA, USA). For the adoptive transfer infection experiments, CD18 KO mice were injected with $1 \times 10^6$ WT and Pcyox1l CRISPRed PMNs IV 6 h prior to infection.

### Sample preparation for LC-MS/MS analysis and data processing

PMNs from infected and noninfected SW and GF SW mice (in quadruplicate) were subjected to in-solution trypsin digestion[56]. Briefly, 200 µl of 8 M Urea (Sigma-Aldrich, U1250, Saint Louis, MO, USA) containing 40 mM HEPES (Sigma-Aldrich, H4034, Saint Louis, MO, USA) was added to the samples, sonicated in a rotating water bath at 4 °C for 15 min (30 s on, 30 s off). The samples were then reduced with 10 mM dithiothreitol (DTT, Sigma-Aldrich, D0632, Saint Louis, MO, USA), alkylated with 55 mM iodoacetamide (IAA, I6125, Sigma-Aldrich, Saint Louis, MO, USA), followed by dilution with 50 mM ammonium bicarbonate (ABC, Thermo Fisher Scientific, A643-500, Waltham, MA, USA) to 2 M urea, and LysC and trypsin (Thermo Fisher Scientific, A40007, Waltham, MA, USA) (protein:enzyme ratio 50:1) and digested overnight. Digestion was stopped by addition of 10% v/v trifluoroacetic acid (TFA, Sigma-Aldrich, 30203, Saint Louis, MO, USA) per sample and the acidified peptides were loaded onto StageTips (containing three layers of $C_{18}$) to desalt and purify. Each sample was divided onto two StageTips (one "working" and one "back-up") and stored at 4 °C until the LC-MS/MS measurement.Samples were eluted from StageTips with 50 µl buffer B 80% acetonitrile (ACN, Thermo Fisher Scientific, TS-51101, Waltham, MA, USA) and 0.5% acetic acid (Thermo Fisher Scientific, A38-212, Waltham, MA, USA), the organic solvent was removed in a SpeedVac concentrator for 20 min, and peptides were resuspended in 10 µl of Buffer A (2% ACN and 0.1% TFA). 3 µl of each sample was analyzed by nanoflow liquid chromatography on an EASY-nLC system (Thermo Fisher Scientific, Bremen, Germany) on-line coupled to an Q Exactive HF-X quadrupole orbitrap mass spectrometer (Thermo Fisher Scientific, Bremen, Germany) through a nanoelectrospray ion source (Thermo Fisher Scientific, Bremen, Germany). A 50 cm column with 75 µm inner diameter was used for the chromatography, in-house packed with 3 µm reversed-phase silica beads (ReproSil-Pur $C_{18}$-AQ, Dr. Maisch GmbH, Germany). Peptides were separated and directly electrosprayed into the mass spectrometer using a linear gradient from 5% to 60% ACN in 0.5% acetic acid over 120 min at a constant flow of 300 nl/min. The linear gradient was followed by a washout with up to 95% ACN to clean the column for the next run. The overall gradient length was 145 min. The QExactive HF-X was operated in a data-dependent mode, switching automatically between one full-scan and subsequent MS/MS scans of the fifteen most abundant peaks (Top15 method), with full-scans ($m/z$ 300–1650) acquired in the Orbitrap analyzer with a resolution of 60,000 at 100 $m/z$.

Raw files were analyzed together using MaxQuant software (version 1.5.6.2)[57]. The derived peak list was searched with the built-in Andromeda search engine[58,59] against the reference Mus musculus proteome downloaded from Uniprot (http://www.uniprot.org/) (53,106 sequences). The parameters were as follows: strict trypsin specificity was required with cleavage at the C-terminal after K or R,

allowing for up to two missed cleavages. The minimum required peptide length was set to seven amino acids. Carbamidomethylation of cysteine was set as a fixed modification (57.021464 Da) and N-acetylation of proteins N termini (42.010565 Da) and oxidation of methionine (15.994915 Da) were set as variable modifications. PSM and protein identifications were filtered using a target-decoy approach at a false discovery rate (FDR) of 1%. 'Match between runs' was enabled with a match time window of 0.7 min and an alignment time window of 20 min. Relative, label-free quantification of proteins was done using the MaxLFQ algorithm[59] integrated into MaxQuant using a minimum ratio count of 2, enabled FastLFQ option, LFQ minimum number of neighbors at 3, and the LFQ average number of neighbors at 6.

## Data analysis

Further analysis of the MaxQuant-processed data was performed using the Perseus software environment (version 1.5.5.5)[60]. The "protein-groups.txt" file was loaded into Perseus. Hits to the reverse database, contaminants, and proteins only identified with modified peptides were eliminated. LFQ intensities were converted to a log scale (log2), and only those proteins which were present in triplicate within at least one sample set were used for further statistical analysis (valid-value filter of 3 in at least one group). Missing values were imputed from a normal distribution (downshift of 1.8 standard deviations and a width of 0.3 standard deviations). The total matrix was imputed using these values, enabling statistical analysis. A Welch's $t$-test was performed to identify proteins with a significant differential expression ($p$-value < 0.05) between baseline and infected SW and C57BL6/N (S0 = 1) samples employing a 5% permutation-based FDR filter. A Principle Component Analysis (PCA) was performed to determine proteome differences at the experiment level, as well as Pearson correlation with hierarchical clustering by Euclidean distance to determine replicate reproducibility. A two-sample Student's $t$-test (S0 = 1) was performed on the entire data set and a 1D-annotation enrichment based on the $t$-test differences with a Benjamini-Hochberg FDR cutoff at 0.05 allowed for visualization of enrichment by keywords within the RStudio platform (http://www.R-project.org/). Specifically, annotation enrichment categories meeting the $p$-value (0.05) and FDR (0.05) cutoffs, along with scores <−0.5 and <0.5 were plotted.

## Purification of PMNs

Murine bone marrow cells were flushed from both hind limbs with PBS (137 mM NaCl, 2.7 mM KCl, 10 mM $Na_2HPO_4$, 1.8 mM $KH_2PO_4$, pH 7.4) supplemented with 2% bovine serum albumin (BSA, American Bioanalytical Inc., 00448-00100, Natick, MA, USA) and 1 mM ethylenediaminetetraacetic acid (EDTA, Invitrogen, AM9260G, Waltham, MA, USA) by a 27-gauge needle (BD, 305136, Franklin Lakes, NJ, USA) and were filtered through a sterile 70-µm nylon cell strainer (Thermo Fisher Scientific, 22363548, Waltham, MA, USA.). Cells were washed and counted with hemocytometer using the trypan blue solution 0.4% (w/v) in PBS. Primary bone marrow neutrophils were purified using the EasySep Mouse Neutrophil Enrichment Kit (StemCell, 19762, Vancouver, Canada).

## Generation and selection of the myeloid knock-out cell lines

**Lentiviral gene transfer**. The ER-Hoxb8 C57BL6/N conditionally immortalized neutrophil progenitor cell lines were established as described in[56]. In general, $0.3 \times 10^5$ C57BL6/N-derived ER-Hoxb8 conditionally immortalized neutrophil progenitors containing Cas9-GFP construct were resuspended in 30 µL of growth medium (RPMI-1640 medium (Corning, 10-040-CV, Corning, NY, USA) supplemented with 10% Fetal Bovine Serum (FBS, Thermo Fisher Scientific, 26140, Waltham, MA, USA), 100 U/ml penicillin/100 µg/mL streptomycin (Thermo Fisher Scientific, 15140-122, Waltham, MA, USA), 2 mM L-Glutamine (Thermo Fisher Scientific, 25030081, Waltham, MA, USA), 2% conditioned CHO media containing Stem Cell Factor (SCF), 0.5 µM

β-estradiol (Sigma-Aldrich, E2758, Saint Louis, MO, USA)) and transduced with lentiviral guides (Supplementary Table 2). Control cells were transduced with a guide to knockdown GFP. The neutrophil progenitors were spinfected at 730 g for 1 h at 15 °C in the presence of polybrene (EMD Millipore, TR-1003-G,, Burlington, MA, USA) (8 µg/ml), then 200 µL of fresh culture medium was added to the cells. After 48 h, 5 µg/mL puromycin (Fisher Scientific, ICN19453910, Waltham, MA, USA) were added to the cultures. The puromycin selection was completed within 72 h, after which the cells were transferred to fresh culture media without puromycin, to expand.

## Cell sorting

After two days of growth, the cells were collected and spun down at 50 g for 5 min at 15 °C, washed with HBSS (HyClone Laboratories Inc, SH3058802, Logan, UT, USA) and sorted. For the single cell sorting, a suspension of $1 \times 10^6$ cells per tube were prepared in 1 mL of HBSS as capture medium. GFP-positive populations were sorted from the transfected clones and non-GFP population was sorted for the control cells (transfected with lentiguides to target GFP) by BD FACSAria cell sorter (BD, Franklin Lakes, NJ, USA). The viability of the cells during the sort was assessed by 7-AAD staining (BioLegend, 420403, San Diego, CA, USA). Single cells were sorted into 96-well plates containing grow media with the total of 2–3 plates per target. After sorting, the plates were maintained at 37 °C humidified incubator with 5% $CO_2$. Two weeks later the sorted clones were transferred into 12-well plates to expand. Each clone was validated by western blotting when antibodies were available or tested for bactericidal activities upon maturation. Selected clones were maintained in freezing media 80% FBS (Thermo Fisher Scientific, 26140, Waltham, MA, USA) and 20% dimethyl sulfoxide (DMSO, Sigma-Aldrich, 472301, Saint Louis, MO, USA) at $1 \times 10^6$ cells per tube. Then the cells were slowly frozen using a cryo-freezing container and stored them at −80 °C.

## Cell lines and treatments

Neutrophil progenitors were maintained in growth media supplemented with 2% conditioned CHO media containing Stem Cell Factor (SCF) and 0.5 µM β-estradiol (Sigma-Aldrich, E2758, Saint Louis, MO, USA). The CRISPRed neutrophil cell lines were authenticated with WB.

HL-60 (ATCC, ccl240) were propagated and differentiated per manufacturer's instructions without authentication.

To differentiate into granulocytes, cells were cultured in RPMI-1640 media supplemented with 10% FBS, 100 U/ml penicillin/100 µg/mL streptomycin, 2 mM L-Glutamine, 2% SCF in the presence or absence of 20 ng/mL GCSF (BioLegend, 574602, San Diego, CA, USA) or GCSF and 5 ng/ml GM-CSF (BioLegend, 576302, San Diego, CA, USA), and maintained for either 3 or 5 days before performing experiments. Cells were seeded into 6-well plates at a density of $2 \times 10^6$ cells/4 mL.

To examine changes in autophagy, neutrophils were treated with 50 µM Chloroquine phosphate (Sigma-Aldrich, 50-63-5, Saint Louis, MO, USA) or vehicle for 4 h before preparing protein lysates. To examine impact of blocking prenylation, neutrophils were treated with 10 µM Cysmethynil (Cayman chemicals, 14745, Ann Arbor, Michigan, USA) for three days.

## Opsonophagocytosis

Opsonophagocytic assays was performed in the presence or absence of opsonic anti-*P.aeruginosa* MoAb (Cam003)[3,4,61]. $1 \times 10^6$ neutrophils were incubated with 10% murine serum, anti-*P.aeruginosa* MoAb (28.6 ng/uL), and *P. aeruginosa* strains 6294 and PAO1 at an MOI of 1:100 for 90 min at 37 °C on a rotator. For murine serum preparation, the fresh blood was left to clot for 1.5 h at room temperature and centrifuged (20 min, 3215 g 10 °C). Aliquots taken at 0 and 90 min, serially diluted, and plated on MacConkey agar (Thermo Fisher Scientific, R01552, Waltham, MA, USA) to determine viable *P. aeruginosa*.

Percent killing was calculated as a ratio of CFU at T90 over the spontaneous bacterial growth in the samples without PMNs or T0.

## ROS measurements

The neutrophils were dispensed in HBSS +/+ (Thermo Fisher Scientific, 14025134, Waltham, MA, USA) into white 96 well plates ($2 \times 10^6$ cells per well) and incubated with 12.5 uM luminol sodium salt (Sigma-Aldrich, A4685, Saint Louis, MO, USA) and 5 U of horseradish peroxidase (Sigma-Aldrich, P8375, Saint Louis, MO, USA) at 37 °C for 5 min. *P.aeruginasa* 6294 was spiked at MOI of 10 in the presence or absence of opsonic anti-*P.aeruginosa* MoAb just before recording the luminescence. Luminescence was measured every 40 s for 30 min duration in SpectraMax L Microplate Reader (Molecular Devices LLC, San Jose, CA, USA) at 470 nm wave length.

## Western blot analysis

Cell lysates were prepared with RIPA buffer (50 mM Tris-HCl (pH 7.4), 150 mM NaCl, 1% NP-40, 0.5% sodium deoxycholate, 0.1% sodium dodecyl sulfate (SDS) and 5 mM EDTA) (Boston BioProducts, BP-115D, Ashland, MA, USA) containing protease inhibitor cocktail (Roche, 11836170001, Basel, Switzerland) at 4 °C. The protein concentration in the lysates was measured using the Bradford method. 20–25 µg of protein was loaded onto NuPAGE 4–12% w/v Bis-Tris 1.5-mm minigel (Invitrogen, NP0335BOX, Carlsbad, CA, USA). Gel electrophoresis was carried out at 200 V in NuPAGE MES SDS (Thermo Fisher Scientific, NP0002, Waltham, MA, USA) or NuPAGE MOPS running buffers (Thermo Fisher Scientific, NP0001) for 90 min for Pcyox1l protein and 35 min for Serpinb1a, LC3, SQSTM1/p62 proteins. The proteins were transferred on polyvinylidene difluoride membranes (Bio-Rad, 162-0261, Hercules, CA, USA) using wet blotting. The mebranes were blocked with 5% non-fat dry milk in Tris-buffered saline solution (TBST,20 mM Tris base, 137 mM NaCl, containing 0.1% Tween 20, pH 7.6) for 1 h and incubated with primary antibodies in TBST containing 5% non-fat dry milk at 4 °C overnight. The membranes were washed using TBST and incubated with the corresponding horseradish peroxidase-conjugated secondary antibodies within 1 h. The chemiluminescence technology (Bio-Rad, ChemiDoc MP Imaging System, Hercules, CA, USA) was utilized for the membrane visualization. The proteins were analyzed by blotting with the following primary antibodies: polyclonal anti-Pcyox1l (Thermo Fisher Scientific, PA5-25523, Waltham, MA, USA; 1:1000), polyclonal anti–Pcyox1l (Biorbyt, orb35904, Durham, NC, USA; 1:500 dilution), anti-Pcyox1 (Santa Cruz, sc-136391, Dallas, TX, USA1:200) polyclonal anti-SerpinB1a (Aviva Systems Biology, OACD05223, San Diego, CA, USA;1:200), polyclonal anti-LC3 (Sigma-Aldrich, L8918, Saint Louis, MO, USA; 1:200), anti-p62 (Sigma-Aldrich, P0067, 1:200), anti-Glyceraldehyde-3-Phosphate Dehydrogenase clone 6C5 (Millipore, MAB374, Burlington, MA, USA,1:500). Goat anti-rabbit IgG–horseradish peroxidase (Santa Cruz Biotechnology, SC-2004, Dallas, TX, USA; 1:2500) and donkey anti-mouse IgG–horseradish peroxidase (Santa Cruz Biotechnology, SC-2314, Dallas, TX, USA; 1:5000) were used as secondary antibodies. PageRuler™ Prestained protein ladder (ThermoFisher, 26616) was used as a standard. In addition to the chemiluminescence, Pcyox1l and GAPDH proteins were visualized with hFAB Rhodamine GAPDH primary antibody (BioRad, 12004167, Hercules, California, USA; 1:1000) and anti-rabbit StarBrightBlue 700 secondary (BioRad, 12004161, Hercules, California, USA; 1:2500). The MW ladders used in the study include Precision Plus protein all blue standard (BioRad, 1610373, Hercules, California, USA), Novex Sharp Pre-Satined Protein Standard (ThermoFisher, LC5800).

## Cytospin staining

Cell suspensions ($0.2 \times 10^5$ cells in 25 µl of PBS per drop) were spun down at 200 rpm for 2 min in Thermo Shandon Cytospin 4 centrifuge (Thermo Shandon, Pittsburgh, PA). The cytospins were stained using Fisher HealthCare™ PROTOCOL™ Hema 3™ Manual Staining System (Thermo Fisher Scientific, 23-123869) per manufacturer's instruction.

## Complete Blood Counts

Complete blood counts (CBCs) performed on an Heska Element HT5 analyzer.

## Flow cytometric characterization of Wild Type and *Pcyox1l* KO mice under homeostatic conditions

Flow cytometric analysis of mature hematopoietic cells was performed from peripheral blood, spleen, and bone marrow samples. We used the following antibodies, including clones, and dilutions: CD11b (M1/70, FITC, 1:400), Ly6G (1A8, APC/Cy7, 1:400), B220 (RA3-6B2, Pacific Blue, 1:200), CD3 (145-2C11, APC, 1:100), CD45 (30-F11, PE/Dazzle 594, 1:400), Ly6C (HK1.4, BV421, 1:200) all purchased from BioLegend. Staining was performed for 45 min at 4 C. Viability was assessed based on 7AAD staining.

Flow cytometric analysis of hematopoietic progenitors was performed from bone marrow samples following ACK hypotonic red blood cell lysis. We used the following antibodies, including clones, and dilutions: CKIT (2B8, PE, 1:100), CD34 (RAM34, FITC, 1:50), SCA1 (D7, BV421, 1:100), CD16/32 (2.4G2, BV605, 1:100), CD48 (HM48-1, APC/Cy7, 1:200), CD150 (TC15-12F12.2, APC, 1:100), CD45 (30-F11, PE/Dazzle 594, 1:400). Lineage positive cells were excluded (B220, CD3, CD4, CD8, CD11b) using a cocktail of biotinylated antibodies (1:50) and a streptavidin-BV650 secondary antibody (1:200). All antibodies were purchased from BioLegend. Primary antibody staining was performed for 60 min at 4 C. Secondary antibody staining was performed for 30 min at 4 C. Viability was assessed based on 7AAD staining.

Flow cytometric data was collected on a BD Celesta Instrument (405 nm, 488 nm, 640 nm laser setup) and analyzed using FlowJo (v10). Graphing and statistical analysis was done using GraphPad Prism (v9). See Supplementary Fig. for gating strategy.

## Flow cytometry analysis of infected corneas

Cornea tissues were dissected from infected animals and processed as in[62]. $1–2 \times 10^6$ cells were resuspended in 50 µL FACS buffer containing Fc block (Biolegend, cat. 101301, clone 93, 1:200). To characterize the frequencies of mature neutrophils, the following markers were used: cKit-FITC (Biolegend, cat. 105805, clone 2B8, dilution 1:200), 7AAD, CD11b-PE (Biolegend, cat. 101207, clone M1/70, 1:200), Ly6C-PE-Cy7 (Biolegend, cat. 128017, clone HK1.4,1:400), CD45-APC (Biolegend, cat. 103111, clone 30F11, 1:400), and Ly6G-BV510 (Biolegend, cat. 127633, clone 1A8, 1:200). All samples were kept in the dark at 4 °C until acquisition where 100,000-500,000 events were acquired for each sample.

## Flow cytometry analysis of in vitro generated myeloid cell lines

400 µL of cells were pelleted and resuspended in 50 µL of FACS buffer (PBS, 5% FBS) containing FC block (CD16/32, Biolgened, cat. 101301, clone 93, 1:200). To quantify immature and mature neutrophils, the following antibodies were used: CD11b PE-CF594 (BD Biosceinces, cat. 562317, clone M1/70, 1:800), CXCR2 PE-Cy7 (Biolegend, cat. 149315, clone SA044G4, 1:200), SiglecF BV421 (Biolegend, cat. 155509, clone S170071, 1:200). 100,000-500,000 events were captured per sample. All flow data was analyzed using Flowjo V 10.7.1.

## Cytokine analysis

Mouse cytokines in corneal lysates were measured using a Meso Scale Discovery (MSD) multiplex 7-spot electrochemiluminescence (ECL) assay by an ultra low noise charge-coupled device (CCD) Imager S600 (Meso Scale Discovery, Gaithersburg, MD, USA). The cytokine array included GM-CSF, IFN-γ, IL-1β, IL-6, TNF-α, VEGF, KC, MIP-2[63,64].

## Metabolic labeling with C15AlkOPP and prenylomic sample processing

Pcyox1L sufficient and deficient myeloid progenitor cell lines and Pcyox1L sufficient and deficient GCSF matured cells were grown in the presence of 10 μM of C15AlkOPP for 24 h[18]. Three replicates of each Pcyox1L deficient and WT for both the matured and progenitor cells were generated. After incubation, cells were harvested, lysed, biotinylated, and precipitated as previously described[18]. Protein pellets were dissolved in 550 μL of 1× PBS/1% SDS and the recovered protein concentration was determined via BCA assay. NeutrAvidin resin (200 μL, Thermo Scientific) was washed with 1× PBS/1% SDS and 0.9–1.1 mg of protein was added, followed by dilution to 2 mg/mL protein with 1× PBS/1% SDS and subsequent binding for 2–3 h at room temperature. Enriched proteins were then washed with 1 mL 1× PBS/1% SDS (3x), then 1 mL 1× PBS (1×), then 1 mL 8 M urea in 50 mM TEAB buffer (3x), and finally with 1 mL 50 mM TEAB buffer (3x). Resin was resuspended in 200 μL of 50 mM TEAB buffer and 1.5 μg of sequencing grade trypsin (Promega Corp) was added. Samples were subjected to on-bead digestion for 24 h at 37 °C. The digestion was stopped with the addition of 2.5 μL of 20% formic acid in water followed by rotating at room temperature for 15 min. Peptides were collected using Pierce spin columns, and the resin was washed with 100 μL of 0.5% formic acid in water, then 100 μL of 30% ACN in water. Collected peptides were dried by lyophilization, then dissolved in 50–70 μL of 100 mM TEAB buffer and quantified with a BCA assay. From each replicate, 10 μg of peptide was supplemented with 150 fmol of internal standard (yeast ADH1, Waters) and labeled with 10 μg of a TMT 6-Ples reagent (Thermo Fisher Scientific) per manufacturer protocol. After 2 h TMT- labeled samples were pooled and solvent was removed with lyophilization. Dried sample was dissolved in 300 μL of 200 mM NH₄HCO₃, pH 10 and loaded onto a in homemade stage tip (three layers of SDB-XC) extraction disks, (3 M, 1.07 mm × 0.50 mm i.d.) in a 200 μL pipette tip. Peptides were then fractionated under high pH reverse phase conditions yielding 7 fractions of 60 μL in volume (5, 10, 15, 20, 22.5, 27.5, 80% ACN in 200 mM NH₄HCO₃, pH 10). The first two fractions were combined and all fractions were dried by lyophilization. Samples were then dissolved in 30 μL of 0.1% formic acid for LC-MS[3] analysis. This method was previously reported[19].

## Prenylomic data acquisition

The peptide fractions underwent a reverse phase separation with a RSLC Ultimate 3000 nano-UHPLC (Dionex) and with a column (75 μm i.d., 45 cm) packed in-house with ProntoSIL C18AQ 3 μm media. Each fraction was separated using a distinct gradient of solvent B (0.1% formic acid in ACN) and solvent A (0.1% formic acid in water) with amounts ranging between 7 and 34% of solvent B for 80 min at 300 nL/min. This was delivered directly into the Orbitrap instrument using a Nanospray Flex source (Thermo Fisher Scientific). For the SPS-MS[3] approach, an Orbitrap Fusion Trihybrid (Thermo Fisher Scientific) mass spectrometer was used. MS[1] scans were collected with the following parameters; 120,000 resolution, max injection time (IT) of 100 ms, range of 320–2000 m/z, and automatic gain control (AGC) target of 200,000. Subsequent data-dependent MS[2] scans were acquired using collision induced dissociation (CID) with these parameters; normalized collision energy (NCE) of 35%, a 1.3 m/z isolation window, a max IT of 100 ms, and an AGC target of 5000. Dynamic exclusion was allowed for 60 s. Acquisition in MS[3] selected the top 10 precursors for fragmentation by high-collisional energy dissociation (HCD) in the orbitrap with the following parameters; NCE of 55%, a 2.5 m/z isolation window, 50,000 AGC target, 120 ms max IT.

## Data analysis and processing

Raw data files were uploaded into MaxQuant (version 1.6.17.0) and searched against a non-redundant mouse data base (UP0000000589) from Uniprot. For all but the following parameters, the default settings were used: Trypsin/P was used for digestion with allowance for 3 missed cleavages, minimum peptide length was set to 7, protein FDR was set to 0.5,.modifications in search oxidation (M) and Acetyl (protein N-Term), and unique and razor peptides were used for quantification. MaxQuant was run through the MaxQuantCmd.exc at the University of Minnesota supercomputing Institute. The protein-group.txt file generated was uploaded into Perseus (version 1.6.14.0). Proteins that were potential contaminantes and reverse were removed by site identification. Raw intensities were transformed to log₂ values, and proteins with more than 3 out of 6 values returning "NaN" after transformation were removed. Missing values were imputed from the normal distribution of the remaining values. Reported values (TMT) were normalized by subtracting rows by mean value and columns by median value. And statistical analysis was performed using a two sample t-test FDR = 5 % and s0 = 0.5.

## Metabolite analysis

Geranylgeranylcysteine and geranylgeranylpyrophosphate quantifications were done by Creative Proteomics.

## Statistical analysis

Statistical analysis of corneal pathology scores, bacterial burden, and cytokine levels were either by Unpaired Student's t-test upon normal distribution, Mann–Whitney U test for pair-wise comparisons, or One-way ANOVA with Dunn's correction for Multigroup comparisons. The analysis of the data from the OPK assays was based on the use of One-way ANOVA. Differences were considered significant if the p value was <0.05 (Prism 9.0 for Macintosh). Multiplex cytokine analysis was conducted by an appropriate Benjamini-Hochberg cut off.

The mice numbers were calculated using Power G analysis software to obtain a significance level p = 0.05 at the 80% probability to find an effect. Typically, cohorts of 7–10 animals per genotype were used depending on availability. Data were excluded from the analysis if animals displayed signs of wounding at a site different from the infected area or ocular clouding at the non-infected eye.

## Reporting summary

Further information on research design is available in the Nature Portfolio Reporting Summary linked to this article.

## Data availability

The PXD031115 submission includes, all Raw files for all proteomic data in Figure 3, peak files, and Maxquant search results. Additionally, for the PXD031115 dataset, a non-redundant uniprot database library, ascension number UP0000000589, was used. This library is included in the PXD031115 deposit. The neutrophil proteomic data have been deposited in the Pride database under accession code PXD009767. This manuscript is accompanied by a Source Data file. Source data are provided with this paper.

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

## Acknowledgements

The authors acknowledge the Minnesota Supercomputing Institute (MSI) at the University of Minnesota for providing resources that contributed to the prenylomic research results reported within this paper (http://www.msi.umn.edu) and Dr. Yingchun Zhao for the assistance in LC-MS[3] analysis in the Mass Spectrometry Core Facility of the Masonic Cancer Center, designated by the National Cancer Institute and supported by P30 CA77598. Authors would like to thank Dr. Alexandra Kulberg (Helois Clinic, Department of Dermatology, Hildesheim, Germany) for the insightful discussions on the project. This work was supported by NIH-NEI RO1 EY022054 to MG, New Frontiers Research Fund: Exploration, Canadian Foundation of Innovation (JELF 38798) to JG. DBS was supported by the American Society of Hematology Scholar Award as well as an NIH K08 (CA201640-01A1). MDD was supported by NIH R35GM141853 and SA was supported by NIH Training Grants T32GM132029 and T32AG029796.

## Author contributions

AK, DQ, OT, SA, CC, AP, JM, RB, DBS, XL, JL and MG performed experiments and analyzed data. JGM generated and analyzed the LC-MS data. AK, AP, DBS, and MG provided insight and generated PMN cell lines. MG, AP, JM, SA generated figures. AP, JGM, SA, DBS, and MG wrote the manuscript. JGM, RF, DBS, SA and MD provided materials, conceptualized experiments, analyzed data, reviewed the manuscript. All authors critically read the manuscript.

## Competing interests

DBS is a co-founder and holds equity in Clear Creek Bio and is a consultant and holds equity in SAFI Biosolutions. The remaining authors declare no competing interests.

## Ethics

All animal experiments were performed following National Institutes of Health guidelines for housing and care of laboratory animals and performed in accordance with institutional regulations after protocol review and approval by BWH IACUC committee and were consistent with the Association for Research in Vision and Ophthalmology guidelines for studies in animals (Protocol number 2018N00002).
