## [Peer Review File · Nature Communications]

Prenylcysteine oxidase 1 like protein is required for neutrophil bactericidal activitiesREVIEWER COMMENTS

Reviewer #1 (Remarks to the Author):

Although there have been extensive studies showing correlation between gut microbiota and immune status, inflammation and infection outcomes in other tissues, the underlying mechanisms are not well understood. In this manuscript, Gadjeva et al provide insight into the role of Pyox1L regulating the metabolic processes that contribute to NADPH oxidase activity and bacterial killing by neutrophils

The authors identified Pyox1L in an unbiased proteomic analysis of neutrophils that was elevated in SPF compared with GF mice. Mass Spec analysis showed a decrease in prenylated proteins and metabolites in Pyox1L^{-/-} neutrophils. the relevance of these findings is shown in Figure 5, where Pyox1L^{-/-} neutrophils have significantly less ROS production in response to infection with *Pseudomonas aeruginosa*. Finally, as ROS is important in bacterial killing, the authors show that in mouse model of *P. aeruginosa* corneal infection, Pyox1L^{-/-} mice exhibit an impaired ability to clear the bacteria, and consequently have more severe disease.

Overall, this manuscript links gut microbiota to the role of Pyox1L mediated prenylation of neutrophil proteins, resulting in a clear phenotype in bacterial infections. The findings are novel, innovative and have broad implications. Overall, this is an outstanding study.

Minor concerns:

Figure 5A needs to be described in more detail. It is also very poor resolution
Several typos and grammatical errors.

Reviewer #2 (Remarks to the Author):

Reviewer's critique

The manuscript written by A. Petenkova et al describes the central role of Pcyox11, a newly discovered protein, in neutrophil antibacterial functions and its regulation by commensal microbes. The study undertakes first a proteomic approach to identify proteins associated with the presence of commensals and then performs a detailed functional analysis of Pcyox11 role in neutrophil effector functions. The manuscript addresses an interesting and novel question on the role of the gut flora in neutrophil maturation and proposes a key player in the process. While the story is interesting, new and the findings mainly support the overall conclusion, I think additional data and manuscript editing would make the paper better. Please see my critiques below:

- Please provide a stronger justification why Pcyox11, not any other of the remaining 9 proteins, was chosen to base the entire manuscript?
- Please describe better how well does the utilized ER-Hoxb8 cell line resemble mature neutrophils? I know the authors cite references for this but it should be more articulated in the current manuscript in the text.
- In figure 5C, the authors should also show the average data of independent experiments to convince the reader about the reproducibility of the experiment. While the representative kinetics indeed show higher ROS output in Wt than Pcyox11 KO PMNs, it is only one experiment, ROS productions has large variability, especially when it comes to a relative luminescent measurement used in the paper
- The text on the Y axis in figure 6D is hard to read. Please fix it
- While the investigators present some data on antimicrobial functions of Pcyox11 ko PMNs,

more could be described in their paper. Are the Pcyox11 KO PMNs deficient in P. aeruginosa phagocytosis as the bioinformatics data suggest?

- Are the PCyox11 KO PMNs deficient in NET formation? The group has all the tools to address this question. NETs represent an important, general antimicrobial mechanism whose usefulness against P. aeruginosa has been questioned by several teams including the investigators.

- While the authors show impaired P. aeruginosa killing by PCyox11 KO PMNs in vitro, they only used one strain. Could they perform the bacterial killing experiment with more P. aeruginosa strains to prove that its effect is not specific to that single bacterium?

- Lipid mediators are also involved in neutrophil chemotaxis. Are the Pcyox11 KO PMNs impaired in their ability to move around? Either in their random migration or in their movement directed by bacteria or bacterial molecules?

- The discussion is very short. The authors should discuss the potential impact and relevance of their findings on Pcyox11 deficiency in more detail.

- Also, the discussion does not mention the potential relevance of the other genes found essential for neutrophil function in their screen. I know they focused on Pcyox11 and will likely work on some of the others in subsequent studies but a paragraph could be added to this current manuscript discussing their potential role and mechanism.

Response to review

Reviewer #1 (Remarks to the Author):

Although there have been extensive studies showing correlation between gut microbiota and immune status, inflammation and infection outcomes in other tissues, the underlying mechanisms are not well understood. In this manuscript, Gadjeva et al provide insight into the role of Pyox1L regulating the metabolic processes that contribute to NADPH oxidase activity and bacterial killing by neutrophils

The authors identified Pyox1L in an unbiased proteomic analysis of neutrophils that was elevated in SPF compared with GF mice. Mass Spec analysis showed a decrease in prenylated proteins and metabolites in Pyox1L^{-/-} neutrophils. the relevance of these findings is shown in Figure 5, where Pyox1L^{-/-} neutrophils have significantly less ROS production in response to infection with *Pseudomonas aeruginosa*. Finally, as ROS is important in bacterial killing, the authors show that in mouse model of *P. aeruginosa* corneal infection, Pyox1L^{-/-} mice exhibit an impaired ability to clear the bacteria, and consequently have more severe disease.

Overall, this manuscript links gut microbiota to the role of Pyox1L mediated prenylation of neutrophil proteins, resulting in a clear phenotype in bacterial infections. The findings are novel, innovative and have broad implications. Overall, this is an outstanding study.

We thank the Reviewer for his kind comments!

Minor concerns:

Figure 5A needs to be described in more detail. It is also very poor resolution –
Corrected.

Several typos and grammatical errors.

Corrected.

Reviewer #2 (Remarks to the Author):

Reviewer's critique

The manuscript written by A. Petenkova et al describes the central role of Pcyox11, a newly discovered protein, in neutrophil antibacterial functions and its regulation by commensal microbes. The study undertakes first a proteomic approach to identify proteins associated with the presence of commensals and then performs a detailed functional analysis of Pcyox11 role in neutrophil effector functions. The manuscript addresses an interesting and novel question on the role of the gut flora in neutrophil maturation and proposes a key player in the process. While the story is interesting, new and the findings mainly support the overall conclusion, I think additional data and manuscript editing would make the paper better. Please see my critiques below:

- Please provide a stronger justification why *Pcyox11*, not any other of the remaining 9 proteins, was chosen to base the entire manuscript?

We chose Pcyox11 based on its predicted metabolic function and high (94%) aminoacid evolutionary conservation among eukaryotes which suggested that the protein has an important biological function. It is noteworthy that this type of evolutionary conservation is rare and is typically present in proteins with fundamental biological significance. For example, the complement system proteins share a high aminoacid homology which reflects evolutionary conservation and similar functions across vertebrate species. We included these comments in the text of the ms p5.

*We provide additional data to show the functional significance of *SerpinB1a*, another protein identified in the proteome data set which we CRISPRed out (Suppl Fig. 5). Importantly, the phenotypes of *Pcyox11* and *SerpinB1a* KO PMNs are comparable therefore we conclude that the pathways are interconnected. We comment on the significance of these findings in the Discussion section of the ms p12.*

- Please describe better how well does the utilized ER-Hoxb8 cell line resemble mature neutrophils? I know the authors cite references for this but it should be more articulated in the current manuscript in the text.

We agree with the Reviewer that expanded explanation would be helpful, but we minimized that information to fit within the requirements for a publication length. Previously, we reported the generation methodology and characteristics for the ER-Hoxb8 C57BL6/N conditionally immortalized neutrophil progenitor cells. These progenitors can be driven to mature to banded and multilobed neutrophils. The maturation can be promoted upon treatment with either GCSF or a combination of GCSF and GM-CSF, traditional PMN driving cytokines. Upon maturation the WT PMNs have: 1) the characteristic for neutrophils banded and multilobed nuclei seen in the cytopins (Suppl. Figure 2 and previous report¹); 2) Comparable opsonophagocytic activities to primary BM PMNs (Fig. 2D, second panel, compare the first, grey bar (BM PMNs) to open bar (WT, cell line derived PMNs). These data are coherent with the previous observations that the in vitro matured WT PMNs form phagocytic vacuoles detectable by electron microscopy (Fig. 6 from ¹).

- In figure 5C, the authors should also show the average data of independent experiments to convince the reader about the reproducibility of the experiment. While the representative kinetics indeed show higher ROS output in Wt than *Pcyox11* KO PMNs, it is only one experiment, ROS productions has large variability, especially when it comes to a relative luminescent measurement used in the paper

*We agree with the reviewer that there is some degree of variability between individual ROS experiments not only in the experiments done by our group, but also by different groups. We provide data from two different experiments with GCSF-matured WT and *Pcyox11* KO cell lines. Cumulatively, these data show that despite the variability, the findings of diminished ROS remain consistent. The in vitro matured *Pcyox11* KO PMNs release less ROS over time.*

Reviewer's questions on ROS

- The text on the Y axis in figure 6D is hard to read. Please fix it
Corrected.

- While the investigators present some data on antimicrobial functions of Pcyox11 ko PMNs, more could be described in their paper. Are the Pcyox11 KO PMNs deficient in *P. aeruginosa* phagocytosis as the bioinformatics data suggest?

We would like to clarify that the presented in the manuscript data are from opsonophagocytic assays following previously established by the lab and published protocols². Data demonstrate significant reductions in the opsonophagocytic properties of Pcyox11 KO PMNs when compared

to WT PMNs confirming the key role for Pcyox11 in mediating neutrophil functionality (Fig 2, panel D, and Supplementary Fig. 3). These assays were performed using the anti-psl MoAb Cam003 to facilitate the opsonophagocytic killing of P. aeruginosa. Our previous experiments showed that in the absence of Cam003, there is no detectable killing by either primary bone marrow derived PMNs or in vitro matured PMN cell lines, therefore the detectable bactericidal capacity of neutrophils is rooted in their opsonophagocytic killing.

- Are the PCyox11 KO PMNs deficient in NET formation? The group has all the tools to address this question. NETs represent an important, general antimicrobial mechanism whose usefulness against P. aeruginosa has been questioned by several teams including the investigators.

The bactericidal capacity of NETs is rather limited with majority of P. aeruginosa clinical isolates been resistant to NET killing³⁻⁵. Recent data show that NETs primary role is to generate a barrier between bacterial biofilms⁶ and the host, however, the issue is controversial as it was reported that NET DNA can be also used to promote biofilm growth. We observed that different strains of P. aeruginosa have distinct binding properties to NETs and different susceptibilities to NET-mediated killing⁴. Because of this we think that an independent study into this subject is needed with dedicated funds to conduct it. Having the topic covered within the current manuscript will dilute out our main message.

- While the authors show impaired P. aeruginosa killing by PCyox11 KO PMNs in vitro, they only used one strain. Could they perform the bacterial killing experiment with more P. aeruginosa strains to prove that its effect is not specific to that single bacterium? *Additional experiments showed that there is no strain-specific bias as opsonophagocytic assays with the laboratory strain PAOI showed similar results to the experiments with the 6294 clinical isolate. Suppl. Fig 3.*

- Lipid mediators are also involved in neutrophil chemotaxis. Are the Pcyox11 KO PMNs impaired in their ability to move around? Either in their random migration or in their movement directed by bacteria or bacterial molecules?

We agree with the Reviewer that this is an important aspect of PMN biology, which is why we examined levels of infiltrating PMNs into infected corneas. We did not observe impairment of PMN recruitment. On the contrary, infected Pcyox11 KO mice had significantly higher absolute numbers of PMNs infiltrating corneas and worse pathology. Cumulatively, we interpret this data as no defects in chemotaxis. We agree that future dedicated experiments should tease out the chemotaxis toward different lipid mediators especially those with key impact on PMN recruitment. Practically, we could not achieve this in the framework of our current research since these experiments will necessitate securing targeted funding for this project, which is currently unavailable.

- The discussion is very short. The authors should discuss the potential impact and relevance of their findings on Pcyox11 deficiency in more detail.

Added.

- Also, the discussion does not mention the potential relevance of the other genes found essential for neutrophil function in their screen. I know they focused on Pcyox11 and will likely work on some of the others in subsequent studies but a paragraph could be added to this current manuscript discussing their potential role and mechanism.

We thank the Reviewer for the important point and have now added more information in the Supplementary figures and Discussion.

- 1 Kugadas, A. *et al.* Frontline Science: Employing enzymatic treatment options for management of ocular biofilm-based infections. *J Leukoc Biol* **105**, 1099-1110, doi:10.1002/JLB.4HI0918-364RR (2019).
- 2 Dwyer, M. & Gadjeva, M. Opsonophagocytic assay. *Methods Mol Biol* **1100**, 373-379, doi:10.1007/978-1-62703-724-2_32 (2014).
- 3 Dwyer, M. *et al.* Cystic fibrosis sputum DNA has NETosis characteristics and neutrophil extracellular trap release is regulated by macrophage migration-inhibitory factor. *J Innate Immun* **6**, 765-779, doi:10.1159/000363242 (2014).
- 4 Shan, Q., Dwyer, M., Rahman, S. & Gadjeva, M. Distinct susceptibilities of corneal *Pseudomonas aeruginosa* clinical isolates to neutrophil extracellular trap-mediated immunity. *Infect Immun* **82**, 4135-4143, doi:10.1128/IAI.02169-14 (2014).
- 5 Young, R. L. *et al.* Neutrophil extracellular trap (NET)-mediated killing of *Pseudomonas aeruginosa*: evidence of acquired resistance within the CF airway, independent of CFTR. *PLoS One* **6**, e23637, doi:10.1371/journal.pone.0023637 (2011).
- 6 Alhede, M. *et al.* The origin of extracellular DNA in bacterial biofilm infections in vivo. *Pathog Dis* **78**, doi:10.1093/femspd/ftaa018 (2020).

REVIEWER COMMENTS

Reviewer #1 (Remarks to the Author):

The authors have adequately addressed all the reviewers' comments

Reviewer #2 (Remarks to the Author):

The authors addressed all my comments or experimental requests in the revised version of the manuscript that has significantly improved.

Reviewer #3 (Remarks to the Author):

While the impact of loss of Pcox11 on neutrophil bactericidal functions looks clear (although will leave it to those in the field to comment on the importance of those findings), linking this phenotype to an impact on protein prenylation and the metabolism of prenylcysteines/isoprenoids in particular (the function of the Pcox enzymes) is a big stretch and is not felt to be currently supported by the data. Specific comments on the study:

1. To date, the Pcox enzymes have only been shown to act on the prenylcysteines (farnesyl-Cys and geranylgeranyl-Cys) produced on the degradation of prenylated proteins. While there may be as-yet-unidentified substrates for these enzymes, there is no reason at present to suspect their loss would impact the prenylated proteins themselves (Fig. 3), or on the levels of FPP and GGPP (Fig. 4B). I suspect this impact is from some secondary aspect of Pcox11 loss, or simply something that happened to the cells on their path to the in vitro analysis. At a minimum, this referee would like to see that putting Pcox11 back into the cells rescues the defects observed.
2. Even more concerning is that the one piece of data that would have been expected from loss of a Pcox enzyme from the cell, an impact on F-Cys and GG-Cys levels. These two compounds would be expected to increase in the KO cells, as an enzyme that degrades them has been removed. Instead, these levels decrease (Fig. 4C), and this is surprisingly interpreted as an outcome that is related to the biology. For reference, a mouse KO of Pcox1 (then referred to as Pcly) published some years ago reported the expected increase in F-Cys and GG-Cys. This paper is cited in the the Discussion (ref. 27), but with the simple statement that there were "...no major in vivo phenotypes" without commenting on the previously reported impact on prenylcysteine levels.

In summary, for the case to be made that the loss of Pcox11 exerts an impact on neutrophil functions via an impact on protein prenylation and/or isoprenoid levels, a plausible explanation of a proposed mechanism underlying this impact would need to be advanced by the authors.

Response to review

We would like to thank Reviewers # 1 and #2 for their feedback and we would like to address Reviewer #3 arguments below:

>While the impact of loss of Pcox11 on neutrophil bactericidal functions looks clear (although will leave it to those in the field to comment on the importance of those findings), linking this phenotype to an impact on protein prenylation and the metabolism of prenylcysteines/isoprenoids in particular (the function of the Pcox enzymes) is a big stretch and is not felt to be currently supported by the data.

>We thank the Reviewer for his comment and would like to highlight the following sets of data that link Pcyox11 deficiency to the mevalonate pathway and, specifically, to protein prenylation:

1. Pcyox11 deficient progenitors and matured neutrophils have significant shifts in the cellular prenylomes (Fig. 3). This is a metabolic labeling experiment which analyzes the selective ability of cells to incorporate exogenous isoprenoid substrate which is incorporated only in the prenylated proteins. This assay demonstrates that the levels of farnesylated and geranylgeranylated proteins are decreased in the absence of Pcyox11.

2. Metabolic profiling experiments:

2a. Measurements of farnesylpyrophosphate and geranylgeranylpyrophosphate in *in vitro* matured PMNs. These metabolites are upstream of the protein prenylation (Fig. 4b). This result highlights the impact of Pcyox11 KO on metabolites in the mevalonate pathway.

2b. Measurements of prenylcysteines in *in vitro* matured cell lines and primary cells. These metabolites are generated upon degradation of prenylated proteins. (Fig. 4c and 4d). These measurements showed a reduction in the levels of these catabolites resulting from the degradation of prenylated proteins.

3. Treatment of WT progenitors with ICMT inhibitor, cysmethynil, which blocks the production of mature prenylated proteins, causes a decrease in the frequencies of matured PMNs, a phenotype similar to that of Pcyox11 KO PMNs (see Fig. 1 below).

4. Our prenylome data points to changes in the prenylation state of several Rabs and Rac proteins in the absence of Pcyox11 (Suppl. Table 3A and Suppl. Table 3B). Previously published data link Rabs with key functional outcomes in neutrophils, namely, neutrophil autophagy, viability, exocytosis, phagocytosis, etc. ^{5, 6,7, 8-11, 12, 13, 14-18} therefore, we feel comfortable stating that the prenylation pathway has a central role in controlling neutrophil biology.

5. Treatment of WT progenitors with cysmethynil, an ICMT competitive inhibitor, that globally blocks the production of the mature functional form of many prenylated proteins, caused a 90% drop in the recovered *in vitro* matured neutrophils (Fig. 1(below)). Notably, the viability of the *in vitro*-matured Pcyox11 KO neutrophils was also significantly decreased despite treatments that typically extend neutrophil viability such as GM-CSF or G-CSF, correlating with changes in the protein prenylation (see ms Suppl. Fig. 5).

Figure 1. Cysmethynil treatment decreases recovery of viable *in vitro* matured PMNs. Two million progenitors were matured with vehicle, GCSF and GCSF/GMCSF treatments in the presence or absence of 10 μ M cysmethynil. Viable cells were counted three days post-maturation. One-way ANOVA comparison.

Cumulatively, our data highlight a connection between Pcyox11 deficiency and a deficiency in the mevalonate pathway leading to reductions in protein prenylation affecting neutrophil responses.

1. To date, the Pcox enzymes have only been shown to act on the prenylcysteines (farnesyl-Cys and geranylgeranyl-Cys) produced on the degradation of prenylated proteins. While there may be as-yet-identified substrates for these enzymes, there is no reason at present to suspect their loss would impact the prenylated proteins themselves (Fig. 3), or on the levels of FPP and GGPP (Fig. 4B). I suspect this impact is from some secondary aspect of Pcox11 loss, or simply something that happened to the cells on their path to the *in vitro* analysis. At a minimum, this referee would like to see that putting Pcox11 back into the cells rescues the defects observed.

We agree that our discovery of the phenotype of the Pcyox11 KO neutrophils is unexpected. We think that both, direct and indirect effects contribute to the phenotype. We ruled out cell line-specific effects as both *in vitro* matured cells and primary BM PMNs from Pcyox11 KO mice share similar phenotypes (Fig. 4C).

We agree that reconstitution experiments would be informative. However, we would like to persuade Reviewer #3 that reconstitution experiments in neutrophils are not feasible presently for several reasons: 1) myeloid progenitor stem cell lines are notoriously difficult to transfect to generate gain-of-function. The current technology is able to transfect only 5% of stem cells including the most recent approaches¹; 2) Prof. S. Catz published that only 5% of mature neutrophils can be transfected which is not enough to get meaningful downstream experiments². 3). Lastly, attempted transfection of neutrophils with RNA carriers or DNA carriers changes their functionality making it difficult to carry out functional assays due to the potent IFN response which is induced by those transfections³. Lentiviral transduction in mature neutrophils has been reported by others who concluded that “*we discourage scientists to use lentiviral transduction methods to manipulate primary mature neutrophils*”⁴. Cumulatively, we can state with significant confidence that reconstitution of mature neutrophils with Pcyox11 is currently challenging. We do plan to approach this question in more amenable cell types, but those studies require establishing the impact of Pcyox11 on functionality of other cell types and are beyond the scope of the current study.

2. Even more concerning is that the one piece of data that would have been expected from loss of a Pcox enzyme from the cell, an impact on F-Cys and GG-Cys levels. These two compounds would be expected to increase in the KO cells, as an enzyme that degrades them has been removed. Instead, these levels decrease (Fig. 4C), and this is surprisingly interpreted as an outcome that is related to the biology. For reference, a mouse KO of Pcox1 (then referred to as Pcly) published some years ago reported the expected increase in F-Cys and GG-Cys. This paper is cited in the the Discussion (ref. 27), but with the simple statement that there were “...no major in vivo phenotypes” without commenting on the previously reported impact on prenylcysteine levels.

The assumption that the new Pcyox11 enzyme acts on prenylcysteines as Pcyox1 is the most likely scenario. Should this be the case, we agree with the Reviewer that we should have seen an increase in the F-Cys or GG-Cys, which is the reason we conducted the metabolomic profiling experiments. However, our data did not support increased F-Cys or GG-Cys levels, but rather decreased levels, which we agree is puzzling. It is possible that the enzyme acts on either full length prenylated proteins or short prenylated peptides. Should this be the case, the prenylome landscape will be changed substantially, causing the remarkable shifts we see. If that is the case, we would not expect to see an increase in the prenylcysteine levels but the exact opposite. We agree that dedicated studies should be carried out to identify the substrates for Pcyox11 but those studies fall outside of the scope of the current work.

We agree with the Reviewer that the phenotype of the Pcyox11 KO PMNs may be indirect due to additional functionalities downstream of the Pcyox11 enzymatic function. We would like to point out that the functional outcomes of the released aldehydes (even for Pcyox1) are currently not established. This highlights how little we know about the prenylation catabolism. Therefore, gaining this knowledge requires dedicated sets of experiments outside the scope of a single paper.

We provide data showing remarkable reductions in the signaling scaffolding protein p62/SQSTM1 (Fig. 2 below and Suppl. Fig. 6 (ms)). This observation is very interesting as the p62 deficiency is associated with mitochondrial dysfunction¹⁹ which causes an overall reduction in the prenylation pathway²⁰. These data point to our claim that Pcyox11 deficiency is linked to diminished prenylation. Where and how Pcyox11 enzyme specificity fits in this scenario deserves a dedicated future study and it is likely that the described decreased prenylation is a consequence of a complex regulation over the pathway.

Figure 2. Pcyox11 matured neutrophils show decreased levels of autophagy marker p62. Pcyox11 deficient and sufficient PMNs were matured with vehicle (V), GCSF (G), and GCSF-GMCSF combination (GG). WB for p62.

Cumulatively, our experimental data strongly support the significant impact of Pcyox11 on cellular prenylation: we show through two independent approaches that cellular prenylomes are shifted in the absence of Pcyox11. Firstly, the cellular prenylome profiles in myeloid progenitors and *in vitro* matured cells are reduced and, secondly, the metabolomic profiling of cell lines and, more importantly, primary bone marrow derived PMNs, demonstrate reductions in the mevalonate pathway metabolites both upstream of the protein prenylation and downstream at the

level of the degradation of the prenylated proteins. Because of these results, we state that the impact of Pcyox11 is significant and with broader consequences than that of Pcyox1. The current work opens-up a broad range of follow-up studies including examining enzyme specificity.

Regarding our citation of the prior work on Pcyox1 (Pcly) (ref. 27), we revised the discussion. Our comments referred to the fact that there is no published “*in vivo*” data showing alterations in the *in vivo* phenotypes such as resistance or susceptibility to infections, autoimmunity, reproduction, etc.

In summary, for the case to be made that the loss of Pcox11 exerts an impact on neutrophil functions via an impact on protein prenylation and/or isoprenoid levels, a plausible explanation of a proposed mechanism underlying this impact would need to be advanced by the authors.

While the identity of Pcyox11 substrates are currently unknown, we provide evidence that Pcyox11 absence causes defects in autophagy exemplified by decreased influx into autophagolysosomal degradation (Suppl. Fig. 5) and diminished p62 levels (Suppl. Fig. 6). It is known that p62 deficiency is associated with mitochondrial defects¹⁹, which translate in reductions of the mevalonate pathway²⁰. Therefore, our data is consistent with what is currently known and represents a breakthrough by demonstrating a connection between Pcyox11, downregulation of prenylation, and reductions in autophagy. Our work prompts many questions: how and which Pcyox11 substrates participate in the process; how this negative feedback regulation is occurring; why p62 levels are reduced etc., but they merit an independent study. Cumulatively, our conclusions are that there is a newly discovered negative feedback mechanism downregulating prenylation and affecting functional outcomes in PMNs.

- 1 Esendagli, G. *et al.* Transfection of myeloid leukaemia cell lines is distinctively regulated by fibronectin substratum. *Cytotechnology* **61**, 45-53, doi:10.1007/s10616-009-9241-9 (2009).
- 2 Johnson, J. L., Ellis, B. A., Munafo, D. B., Brzezinska, A. A. & Catz, S. D. Gene transfer and expression in human neutrophils. The phox homology domain of p47phox translocates to the plasma membrane but not to the membrane of mature phagosomes. *BMC Immunol* **7**, 28, doi:10.1186/1471-2172-7-28 (2006).
- 3 Tamassia, N. *et al.* IFN-beta expression is directly activated in human neutrophils transfected with plasmid DNA and is further increased via TLR-4-mediated signaling. *J Immunol* **189**, 1500-1509, doi:10.4049/jimmunol.1102985 (2012).
- 4 Geering, B., Schmidt-Mende, J., Federzoni, E., Stoeckle, C. & Simon, H. U. Protein overexpression following lentiviral infection of primary mature neutrophils is due to pseudotransduction. *J Immunol Methods* **373**, 209-218, doi:10.1016/j.jim.2011.08.024 (2011).
- 5 Mochizuki, Y. *et al.* Phosphatidylinositol 3-phosphatase myotubularin-related protein 6 (MTMR6) is regulated by small GTPase Rab1B in the early secretory and autophagic pathways. *J Biol Chem* **288**, 1009-1021, doi:10.1074/jbc.M112.395087 (2013).

- 6 Ravikumar, B., Imarisio, S., Sarkar, S., O'Kane, C. J. & Rubinsztein, D. C. Rab5 modulates aggregation and toxicity of mutant huntingtin through macroautophagy in cell and fly models of Huntington disease. *J Cell Sci* **121**, 1649-1660, doi:10.1242/jcs.025726 (2008).
- 7 Bridges, D. *et al.* Rab5 proteins regulate activation and localization of target of rapamycin complex 1. *J Biol Chem* **287**, 20913-20921, doi:10.1074/jbc.M111.334060 (2012).
- 8 Longatti, A. *et al.* TBC1D14 regulates autophagosome formation via Rab11- and ULK1-positive recycling endosomes. *J Cell Biol* **197**, 659-675, doi:10.1083/jcb.201111079 (2012).
- 9 Longatti, A. & Tooze, S. A. Recycling endosomes contribute to autophagosome formation. *Autophagy* **8**, 1682-1683, doi:10.4161/auto.21486 (2012).
- 10 Puri, C. *et al.* The RAB11A-Positive Compartment Is a Primary Platform for Autophagosome Assembly Mediated by WIPI2 Recognition of PI3P-RAB11A. *Dev Cell* **45**, 114-131 e118, doi:10.1016/j.devcel.2018.03.008 (2018).
- 11 Puri, C., Vicinanza, M. & Rubinsztein, D. C. Phagophores evolve from recycling endosomes. *Autophagy* **14**, 1475-1477, doi:10.1080/15548627.2018.1482148 (2018).
- 12 Hirota, Y. & Tanaka, Y. A small GTPase, human Rab32, is required for the formation of autophagic vacuoles under basal conditions. *Cell Mol Life Sci* **66**, 2913-2932, doi:10.1007/s00018-009-0080-9 (2009).
- 13 Jager, S. *et al.* Role for Rab7 in maturation of late autophagic vacuoles. *J Cell Sci* **117**, 4837-4848, doi:10.1242/jcs.01370 (2004).
- 14 Mauvezin, C. *et al.* Coordination of autophagosome-lysosome fusion and transport by a Klp98A-Rab14 complex in *Drosophila*. *J Cell Sci* **129**, 971-982, doi:10.1242/jcs.175224 (2016).
- 15 Johnson, J. L., Pestonjamas, K., Kiosses, W. B. & Catz, S. D. Super-Resolution Microscopy and Particle-Tracking Approaches for the Study of Vesicular Trafficking in Primary Neutrophils. *Methods Mol Biol* **2233**, 193-202, doi:10.1007/978-1-0716-1044-2_13 (2021).
- 16 Johnson, J. L., Hong, H., Monfregola, J. & Catz, S. D. Increased survival and reduced neutrophil infiltration of the liver in Rab27a- but not Munc13-4-deficient mice in lipopolysaccharide-induced systemic inflammation. *Infect Immun* **79**, 3607-3618, doi:10.1128/IAI.05043-11 (2011).
- 17 Johnson, J. L. *et al.* Identification of Neutrophil Exocytosis Inhibitors (Nexinhibs), Small Molecule Inhibitors of Neutrophil Exocytosis and Inflammation: DRUGGABILITY OF THE SMALL GTPase Rab27a. *J Biol Chem* **291**, 25965-25982, doi:10.1074/jbc.M116.741884 (2016).
- 18 Ramadass, M., Johnson, J. L. & Catz, S. D. Rab27a regulates GM-CSF-dependent priming of neutrophil exocytosis. *J Leukoc Biol* **101**, 693-702, doi:10.1189/jlb.3AB0416-189RR (2017).
- 19 Seibenhener, M. L. *et al.* A role for sequestosome 1/p62 in mitochondrial dynamics, import and genome integrity. *Biochim Biophys Acta* **1833**, 452-459, doi:10.1016/j.bbamcr.2012.11.004 (2013).

- 20 Wall, C. T. J. *et al.* Mitochondrial respiratory chain dysfunction alters ER sterol sensing and mevalonate pathway activity. *J Biol Chem* **298**, 101652, doi:10.1016/j.jbc.2022.101652 (2022).

REVIEWERS' COMMENTS

Reviewer #1 (Remarks to the Author):

The authors did an outstanding job in responding to reviewers' questions and concerns. However, although there are newly highlighted areas in the revised manuscript, it is not clear how many of the points raised by the reviewer are addressed in the text.

Reviewer #2 (Remarks to the Author):

The authors responded to the important critiques of reviewer 3 in their revision. The main findings of this paper are the investigation of a previously unknown protein: prenylcysteine oxidase 1 like (Pcyox1l) in neutrophil functions. Deficiency in Pcyox1l leads to significant reductions in metabolites from the mevalonate pathway and de novo protein prenylation that affect the functions of the NADPH oxidase and autophagy. Pcyox1l KO mice are more susceptible to infection with *P. aeruginosa*. The authors identify a function of Pcyox1l as a basic regulator of the prenylation pathway and modulator of neutrophil responses.

In my view, this work is original and provides new information compared to the established literature. Most of the presented results support the authors' conclusions. While important points have been raised by reviewer 3 regarding a lack of a causative link between the neutrophil phenotypes and protein prenylation, I agree that some of these questions are very difficult to address in the unique cell type of neutrophils. In response to these critiques, the authors changed their manuscript, specially the discussion, significantly acknowledging these limitations.

The methodology is sound, the work uses methods widely accepted in neutrophil research. Sufficient details are provided in the methods to enable reproduction of this work by other investigators.

Response to review

We would like to thank the Editor, Editorial Board, and the Reviewers for providing us with this unique opportunity to discuss our science and improve its presentation. Additionally, we would like to thank the Editor and the Reviewers for their time spent on reading this submission, the thought-provoking and in-depth discussion. We think that the process allowed us to reflect, review, and revise the manuscript which ultimately strengthened our presentation. It was an outstanding learning experience. Thank you!

A discussion point stating the difficulties and experimental limitations that currently prevent the modification for putting *Pcox11* back into the neutrophils, please include sufficient references to support your statement and a disclaimer.

>Statements relating to the technical challenges in performing reconstitution experiments in neutrophils are added to the Discussion section of the manuscript (p13).

We also wish this point to be included in the discussion on this finding (whether the effect is direct or indirect) which would suitably acknowledge this point of the reviewer.

>We commented on the potential mechanisms explaining the overall inhibition of the mevalonate pathway under homeostatic conditions in the Discussion section (p12-p13).

Please ensure the additional data and discussion in response to the concerns on the expected decrease in F-Cys and GC-Cys are maintained in your manuscript and fully discussed.

> We describe: 1) the relative levels of GGPP and FPP in vehicle and GCSF *in vitro* matured *Pcyox11* CRISPRed and WT PMNs (Fig 4b); 2) relative levels of geranylgeranylecysteine and farnesylecysteine levels in vehicle and GCSF-*in vitro* matured *Pcyox11* CRISPRed and WT PMNs (Fig. 4c) and a comparable phenotype in primary BM-derived neutrophils. Cumulatively, data demonstrate reductions in these metabolites in the absence of *Pcyox11* under homeostatic conditions (Fig. 4d).

>We also provide additional new data from BM-derived primary PMNs isolated from infected mice. In contrast to the homeostatic state, the relative GG-Cys levels were significantly increased in the *Pcyox11* KO PMNs when compared to WT-derived PMNs. These data support the notion that *Pcyox11* catabolizes prenylecysteines during infection.

>Cumulatively, our data highlight that the *Pcyox11* deficiency distinctly impacts non-activated and infection-activated PMNs. We suggest that the steady state phenotype is likely an indirect consequence of the *Pcyox11* loss through an impact on mitophagy/autophagy. This indirect effect does not negate the enzymatic role of *Pcyox11*. On the contrary, our data point to homeostatic adaptation to the loss of enzyme function by downregulation of the prenylation pathway influx. During infection, changes in the BM-derived PMNs are consistent with the expected enzyme function.

The reviewer also found your conclusions that *Pcox11* loss alters neutrophil function and this is due to prenylation preliminary. Whilst I agree experimentally addressing this is beyond the scope

of the current manuscript we expect you to acknowledge this as a limitation of the study and include suitable discussion.

>We acknowledge a limitation in our experiments, namely not being able to perform reconstitution experiments with active or inactive Pcyox11 into neutrophils. We added statements to this effect on p13. Transfection of mature neutrophils is experimentally challenging. Our findings constitute a good foundation for follow-up experiments in other cell types which are more amenable to reconstitution experiments.

Reviewer #1 (Remarks to the Author):

The authors did an outstanding job in responding to reviewers' questions and concerns. However, although there are newly highlighted areas in the revised manuscript, it is not clear how many of the points raised by the reviewer are addressed in the text.

> We thank the Reviewer for this comment. We reorganized the Discussion section of the manuscript by adding comments to explain why we think the prenylation pathway is reduced under homeostatic conditions and acknowledge our limitations in directly examining the impact of Pcyox11 enzyme activity by reconstituting *Pcyox11* KO PMNs.

Reviewer #2 (Remarks to the Author):

The authors responded to the important critiques of reviewer 3 in their revision. The main findings of this paper are the investigation of a previously unknown protein: prenylcysteine oxidase 1 like (Pcyox11)in neutrophil functions. Deficiency in Pcyox11 leads to significant reductions in metabolites from the mevalonate pathway and de novo protein prenylation that affect the functions of the NADPH oxidase and autophagy. Pcyox11 KO mice are more susceptible to infection with *P. aeruginosa*. The authors identify a function of Pcyox11 as a basic regulator of the prenylation pathway and modulator of neutrophil responses.

In my view, this work is original and provides new information compared to the established literature. Most of the presented results support the authors' conclusions. While important points have been raised by reviewer 3 regarding a lack of a causative link between the neutrophil phenotypes and protein prenylation, I agree that some of these questions are very difficult to address in the unique cell type of neutrophils. In response to these critiques, the authors changed their manuscript, specially the discussion, significantly acknowledging these limitations. The methodology is sound, the work uses methods widely accepted in neutrophil research. Sufficient details are provided in the methods to enable reproduction of this work by other investigators.

>We thank the Reviewer for his support of the submitted manuscript.

Additional minor comments:

We made the following changes to the Figures:

1. Fig. 1c was revised. We provided additional data to Fig. 1c, recalculated, and updated analysis.

2. Fig. 2: Added new Fig. 2c representing predicted structures for huPcyox11 and muPcyox11 indicating domain similarity.
3. Fig. 2d: The representative WB image demonstrating the loss of Pcyox11 band in the *Pcyox11* CRISPRed clones was substituted with a new image, showing ladder, Pcyox11 and GAPDH staining simultaneously, consistent with the Journal's requirements for submitted images. This change was necessitated as our old WB data lacked recognizable ladder since the membranes were stripped multiple times before GAPDH imaging.
4. We expanded Fig. 2 by adding a new Suppl. Fig. 2 with additional WB-derived data demonstrating loss of Pcyox11 in the other two *Pcyox11* CRISPRed clones (clones 1 and 3) (Suppl. Fig, 2a). We also included additional WB data showing Pcyox11 profiles in the *in vitro*-matured CRISPRed clones PMNs (Suppl. Fig 2b).
5. Fig. 4 has been revised to include new metabolite quantification data. These data are very exciting as they demonstrated that the inhibition of the mevalonate pathway occurs in the absence of Pcyox11 in primary BM-derived PMNs under homeostatic conditions, whereas during infection, the primary BM-derived PMNs show the expected build-up of substrate, consistent with the comments made by Reviewer 3.
6. The old Fig. 5 data is moved to Supplementary information. Data from that figure are split between new Suppl. Fig. 4 and new Suppl. Fig 5.
7. The new Suppl. Fig. 4a shows changes in ROS in the presence or absence of anti-P. aeruginosa reactive antibody. This is expanded data on ROS release which we referred to in the previous submission as 'data not shown'.
8. The new Suppl. Fig. 6 shows WB data demonstrating presence of Pcyox1 in the *Pcyox11* KO BM-derived PMNs. These data were previously referred to as 'data not shown'. Given that the build-up of geranylgeranylecysteine metabolites is occurring in PMNs derived from the infected *Pcyox11* KO mice despite the presence of the established prenylcystinylase Pcyox1 in these cells, we suggest that Pcyox11 is a key enzyme in the mevalonate pathway
9. The new Fig. 6a shows additional information on peripheral blood leukocyte frequencies in infected Pcyox11 and WT mice, consistent with our previous statements of altered PMN functionality rather than trafficking. This change was not requested but we opted to include the data as it supports our conclusions. The legend was also revised to rectify inconsistencies in data description.
10. New Fig. 7.: We revised the model figure in hopes for improved presentation.